# Soil bacterial diversity mediated by microscale aqueous-phase processes across biomes

Samuel Bickel [1]* & Dani Or [1,2]

Soil bacterial diversity varies across biomes with potential impacts on soil ecological functioning. Here, we incorporate key factors that affect soil bacterial abundance and diversity across spatial scales into a mechanistic modeling framework considering soil type, carbon inputs and climate towards predicting soil bacterial diversity. The soil aqueous-phase content and connectivity exert strong influence on bacterial diversity for each soil type and rainfall pattern. Biome-specific carbon inputs deduced from net primary productivity provide constraints on soil bacterial abundance independent from diversity. The proposed heuristic model captures observed global trends of bacterial diversity in good agreement with predictions by an individual-based mechanistic model. Bacterial diversity is highest at intermediate water contents where the aqueous phase forms numerous disconnected habitats and soil carrying capacity determines level of occupancy. The framework delineates global soil bacterial diversity hotspots; located mainly in climatic transition zones that are sensitive to potential climate and land use changes.

---

[1] Soil and Terrestrial Environmental Physics (STEP), Department of Environmental Systems Sciences (USYS), ETH Zürich, Zürich 8092, Switzerland.
[2] Division of Hydrologic Sciences, Desert Research Institute, Reno, NV, USA. *email: samuel.bickel@usys.ethz.ch

Soil hosts unparalleled bacterial diversity, ranking highest among all other compartments of the biosphere[1–3]. The number of bacterial phylotypes ranges between $10^2$ and $10^6$ per gram of soil[1,2,4], with high values similar to the diversity in all of earths environments[3]. This immense richness is often attributed to soil's intrinsically heterogeneous physical and chemical micro-environments[5–9]. The complex structure of soil pores offers numerous refugia for hosting diverse bacterial species[9]. This wide range of microhabitats is particularly important for maintaining the rare components of the soil microbiome. Low abundance bacterial species play important roles in key biogeochemical processes[10,11] and serve as a "seed bank" for species richness[12]. Microbial diversity is manifested both at the scale of soil grains[8] and at very large scales across climatic regions and terrestrial biomes[2,13,14]. These observations often include variations in microbial biomass that responds to resource availability and affects bacterial diversity at all scales[15–17]. For example, well-established observations of microbial abundance variations with soil depth[18] could confound inferences of bacterial richness by promoting the detection of low abundant species in resource-rich environments.

Quantifying the roles of soil factors, such as soil texture, porosity and hydration conditions in relation to climate and vegetation cover, is an important step towards disentangling bacterial diversity and abundance as suggested by recent empirical evidence[17]. Soil chemical properties such as pH[2,14,17,19] and organic carbon content[15–17] together with climatic attributes, such as aridity index[15], precipitation[2,17] and temperature[13], have been identified as important explanatory variables. Yet, the rapid expansion of soil bacterial diversity datasets has not been met with similar development of predictive models for interpretation of the observed spatial patterns[20]. Improved predictability of soil bacterial diversity could be essential for understanding soil bacterial functioning; from contributions to soil respiration[11,21] to the resistance of bacterial communities to invasion by pathogens[22].

Such endeavors invariably require development of mechanistic frameworks for systematic incorporation of the various factors that affect soil bacterial diversity. In this study, we capitalize on recent empirical[2,8,13,15,17,23] and theoretical developments[7,24,25] to generalize the role of soil aqueous microhabitat fragmentation and its nearly universal role in mediating bacterial diversity across soil types and climatic conditions. To characterize the average conditions in soils and facilitate long-term predictions, we define a soil climatic water content that combines rainfall patterns and volumetric soil water holding capacity into a well-defined attribute. This measure considers the average duration between soil wetting events important for diversity maintenance (see Methods). Under a wide range of climatic conditions, soils remain unsaturated with the bacterial aqueous habitats fragmented to varying degrees based on soil type and rainfall dynamics (amount and frequency). A critical hypothesis is that the microscale arrangement of water retained in soil pores defines the size distribution and connectedness of aqueous bacterial habitats that, in turn, affect diffusion rates of substrates, the rates and spatial extents of cell motility[25,26] and opportunities for cell-to-cell interactions[27]. The objective of this study was to formalize the influence of these abiotic factors in a heuristic framework that enables quantitative representation of soil bacterial abundance and diversity at scales ranging from grains to watersheds and beyond.

The core of the model is the quantification of numbers and sizes of aqueous bacterial habitats considering climatic water contents and soil types. We use concepts of percolation theory to describe the size distribution of aqueous patches[24] that could support bacterial cells. Soil organic carbon input flux, derived from the net primary productivity (NPP), and mean annual temperature (MAT) are used to estimate a soil-carrying capacity that defines limits for the abundance of bacterial cells (Fig. 1). For simplicity, we first assume that each isolated aqueous patch is inhabited by a single bacterial phylotype (hereafter referred to as "species"). This heuristically enables estimation of bacterial diversity based on the species abundance distribution (SAD) deduced from the size and number distribution of microscale aqueous habitats. The framework expresses soil bacterial diversity at two interlinked spatial scales: at the single aqueous habitat scale and at the soil sample scale that can contain many isolated aqueous habitats.

Modeled trends of soil bacterial carrying capacity and diversity are compared to empirical observations[1,4,18] across terrestrial biomes and suggest a peak in bacterial diversity at intermediate climatic water contents. To evaluate predictions by this aqueous-phase fragmentation-based heuristic model (HM), we employ a detailed, spatially explicit individual-based model (SIM) that mechanistically simulates bacterial communities growing on hydrated soil surfaces[7,25]. The SIM enables systematic variations of hydration conditions and tracks the growth and life history of multiple species interacting on soil grain surfaces (see Methods).

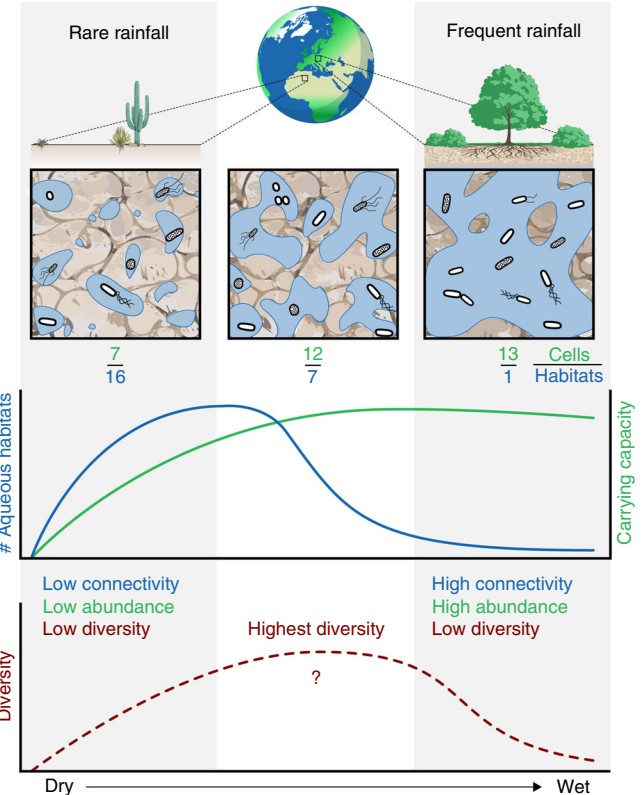

**Fig. 1 Illustration of aqueous habitat fragmentation and carrying capacity in relation to climatic water contents.** In regions where rainfall is frequent, the soil aqueous phase is largely connected and provides a common habitat for cells of different bacterial species. In soils of drier regions, the aqueous phase is increasingly fragmented and offers a large number of distinct habitats. When the soil becomes sufficiently dry, almost all aqueous habitats are physically isolated and might contain only a few species. Additionally, the total number of cells that can be maintained (potential carrying capacity) is reduced and smaller patches become uninhabited. The specific carrying capacity in a biome is based on carbon input flux and temperature that establish an upper bound on bacterial cell density (rarely realized in any particular location due to other limiting factors). The numbers below each panel indicate the number of cells per number of habitats. Diversity is expected to drop in dry regions with low cell abundance and in wet regions with high habitat connectivity.

The simple HM does not differentiate between the roles of legacy and environmental conditions in shaping soil bacterial diversity. As evidenced from the choice of climatic averaging and the implicit representation of species with no taxonomic attribution, the focus lies on the role of aqueous habitats and their average connectivity. Other factors at play such as soil chemistry and the presence of larger organisms are not modeled. We refer to "microbes" for aspects that apply to all microbial life in soil (bacteria, fungi, protists and viruses), and specifically to bacteria for modeling and quantification of diversity and abundance. Summarizing, we propose a hydration-centered modeling framework that considers the interplay of climatic water content; carbon input flux and temperature in shaping soil microhabitats and thus bacterial diversity.

## Results

**Estimation of soil bacterial carrying capacity**. We evaluated theoretical estimates of soil bacterial carrying capacity using previously published measurements of soil microbial carbon[18]. The HM assumes that a certain proportion of the annual NPP-derived organic carbon input is allocated to bacteria (24% of NPP for bacterial respiration[28,29]). We found that varying the range of expected values (14–30% of NPP[28]) had little impact on estimates of carrying capacity. A constant value of this respiratory fraction was therefore considered based on mechanistic model simulations[28]. We employ a basic estimate of bacterial cell maintenance rate of 1.5 $gC\,gC_{cell}^{-1}\,y^{-1}$ ($\approx 10^{-4}\,gC\,gC_{cell}^{-1}\,h^{-1}$) and adjust it according to the local mean annual temperature (MAT)[30] to account for different climatic regions. Combining local annual NPP and adjusted cell maintenance rate, we derive estimates of soil bacterial carrying capacity as upper bounds for soil bacterial cell density (Fig. 2a). Despite the many simplifying assumptions, we obtain reasonable estimates of potential soil bacterial carrying capacity that are comparable with observations of realized bacterial cell density across a range of environmental conditions. Model estimates of soil-carrying capacity for three values of MAT are depicted in Fig. 2a (representing the median of three groups: ≤0 °C, 0−15 °C and >15 °C with −2, 9 and 19 °C, respectively). Observed cell densities tend to be higher for colder regions as considered by the HM. We note that soil bacterial cell density is expected to vary with soil depth due to the distribution of organic carbon flux from the soil surface and distribution by plant roots[18]. Soil bacterial carrying capacity decreases steeply with depth and was represented parametrically by a lognormal distribution ($\mu = 0.18$, $\sigma = 1.00$) (Fig. 2b). The lognormal distribution provided a better global representation of the average topsoil carrying capacity (upper 10 cm, Supplementary Fig. 1) over the previously reported exponential model[18]. It is important to keep in mind that the estimated soil-carrying capacity was defined independently from bacterial diversity and values were calculated globally based on NPP, MAT and soil depth.

**Modeling bacterial diversity considering climate and soil**. The simple HM was developed in two conceptual steps. We first assumed only a single species per aqueous habitat. This approach, although useful as a heuristic, exhibited some limitations for large aqueous habitats under wet conditions (see comparison of species abundance distributions below). We thus adapted the model to allow multiple species in large habitats by assigning the number of species $N_{sp}$ proportional to the length scale of a habitat of size $s$ ($N_{sp} \sim s^{1/d}$, $d = 2$ or $3$ = dimensionality). Hence, the HM links species richness to the soil aqueous-phase fragmentation via percolation theory and accommodates the possibility of multiple species per habitat. For most unsaturated conditions the refined formulation does not alter the prediction since small habitats are

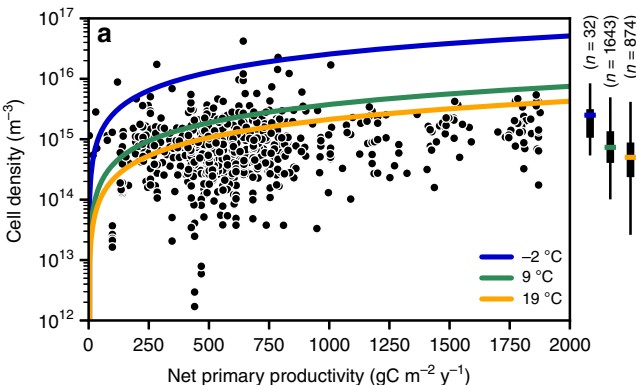

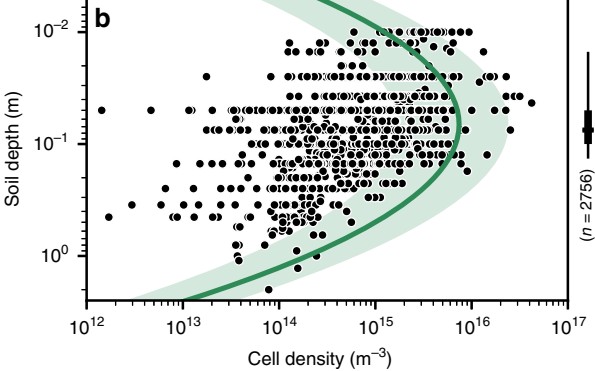

**Fig. 2 Soil bacterial abundance varies in relation to carbon input, temperature and soil depth. a** Bacterial cell density at soil-carrying capacity as a function of net primary productivity (NPP) with model estimates sensitive to mean annual temperature (MAT) (solid lines). Estimates are compared with measured data of microbial biomass[18] converted to bacterial cell density and are grouped by temperature (MAT ≤ 0 °C, 0 °C > MAT ≤ 15 °C, MAT > 15 °C). Each group's median is reported in the figure legend in blue, green and orange, respectively. The distributions of cell densities are indicated for each temperature group as the central 50 and 95% range. **b** Variations of bacterial cell density with soil depth. The lognormal fit provides bounds on cell density (carrying capacity) for intermediate MAT (solid line) and for the central 95% of NPP (shaded area). Observed estimates of cell density are reported for their average sampling depth. Most samples were taken above 10 cm as shown in the boxplot. Source data are provided as a Source Data file.

likely to host only a single species. In the following we refer to the multispecies HM if not stated otherwise. We have used median values of global soil-carrying capacity to describe trends in soil bacterial diversity across soil types and climatic regions. Comparisons of model estimates with empirical observations of bacterial richness obtained from the studies of Thompson et al. (EMP)[1] and Delgado-Baquerizo et al. (DEL)[4] are depicted in Fig. 3 along with the mechanistic predictions by the SIM. We have expressed mean soil hydration status via the climatic water content that is a proxy for average soil wetness and habitat connectivity. Soil and climatic variables were compiled from different sources (Supplementary Table 1) with matched geographical coordinates and soil depths for the samples. We present soil bacterial richness (total number of types) and note that taxonomic assignment was absent for the phylotypes detected in EMP. Bacterial richness was binned by water contents because some hydration conditions were over-represented (bin width: 0.05). Since richness in the EMP data was measured at different soil depths, they were also grouped to top and subsoil (<25 cm and ≥25 cm). Exact number of samples per group are reported in Supplementary Table 2. The EMP data

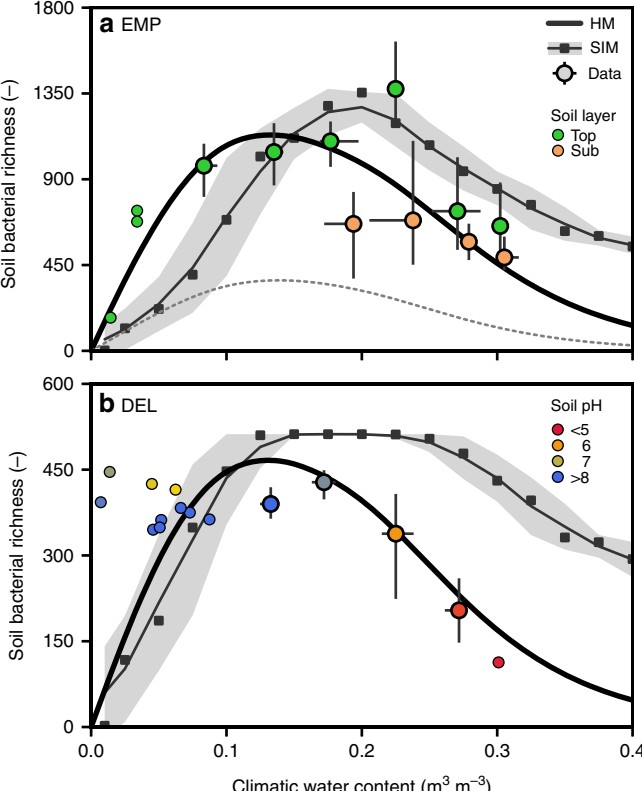

**Fig. 3 Observed and predicted variations in soil bacterial diversity with climatic water content. a, b** Estimates of bacterial richness from two studies are binned by climatic water contents (bin width: 0.05) and the median and interquartile range are reported (circles and bars, respectively). The exact number of samples per group is listed in Supplementary Table 2. Individual data points are shown for bins containing less than ten samples (small circles). The solid black lines correspond to predictions by the fragmentation-based heuristic model (HM) for median carrying capacity specific to each dataset. The square symbols, thin solid line and shading (mean, rolling mean ± SD, $n = 12$) depict simulated bacterial richness using the spatially explicit individual-based model (SIM) for different water contents. **a** Bacterial richness from the Earth Microbiome Project (Thompson et al.—EMP)[1] was reported for different soil depths and thus grouped accordingly (<25 and ≥25 cm, top- and subsoil, respectively). The dashed line represents a model scenario with reduced carrying capacity by considering only the subsoil. **b** Soil bacterial richness from a recent study (Delgado-Baquerizo et al.—DEL)[4]. Colors indicate reported soil pH, which has been shown to be affected by climate[36]. For comparison with the DEL dataset, only the top 512 species were considered in the SIM. Source data are provided as a Source Data file.

display a tendency of lower values of richness in the subsoil (Fig. 3a). In the DEL dataset, measurements were taken at the same soil depth, and soil pH is reported instead (Fig. 3b). We observe a strong tendency of lower soil pH in climatically wetter soils. The results depict an average decrease in bacterial richness where the soil becomes saturated as also predicted by the HM for median soil-carrying capacity (Fig. 3a, b). The modeled sensitivity to soil-carrying capacity is shown for a scenario of reduced cell densities (e.g. less carbon input to deeper soil layers; Fig. 3a dashed line). We emphasize that parameters were not fitted to observed diversity data, but rather are based on mean values for soil properties (porosity $\theta_s = 0.49$ and 0.47; sample length $L = 5$ and 6 mm; textural length $\delta = 0.07$ and 0.1 mm; for EMP and DEL, respectively). Additionally, we used a fixed value for the critical water content ($\theta_c \approx 0.15$) and a threshold for the number of cells $N_{cell}$ needed to

model occupancy of potential habitats ($N_{cell} > 4000$). Lastly, we compared the aqueous-phase fragmentation-based HM to numerical simulations of the SIM. We simulated the spatially explicit growth and movement of individual cells in a diverse bacterial community on heterogeneous soil pore surfaces. Qualitatively, both HM and SIM predict similar trends of variations in bacterial richness with soil hydration conditions as estimated from the EMP and DEL datasets (Fig. 3a, b). In addition to removing single cells (singletons) from the simulated communities, the modeled species counts were rarefied to 5000 and 1000 for comparison with EMP and DEL, respectively. To compare with the DEL dataset, simulated bacterial richness is reported only for the 512 most abundant species and describes the observed invariance of richness towards low climatic water contents (Fig. 3b). The discrepancy in water contents where richness peaks (between HM and SIM) is attributed to the dimensionality of the models (three for HM, two for SIM) and is well captured by the percolation-based HM in two dimensions (Supplementary Fig. 2).

**Species abundance distribution varies with hydration status.** We quantified variations in bacterial species abundance distribution (SAD) with soil attributes and climatic water contents in comparison with empirical estimates from the EMP and DEL datasets (Supplementary Fig. 3). Here we used soil properties and carrying capacity specific for each geographical location and soil depth. The results show good alignment of the single-species model predictions with observed relative SADs and resulted in Pearson correlation values of 0.84 ($n = 230$) and 0.76 ($n = 218$) for the EMP and DEL datasets, respectively (Supplementary Fig. 3a, b). Nevertheless, the single-species HM erroneously predicts a higher proportion of the most abundant species than observed. We attribute this systematic overestimation to the stringent assumption of one single species per aqueous (micro-) habitat. This discrepancy suggests that the single species per aqueous habitat assumption may not hold for very large aqueous habitats in wet soil that could host multiple species. To rectify this limitation, we considered a scenario where the number of species $N_{sp}$ is assumed proportional to the size $s$ of an aqueous habitat ($N_{sp} \sim s^{1/3}$). This relaxed occupancy assumption improved Pearson correlations to values of 0.88 ($n = 230$) and 0.84 ($n = 218$) for the EMP and DEL datasets, respectively (Supplementary Fig. 3c, d). Predictions by the HM for ranked SADs compare qualitatively with observations that were grouped by average hydration conditions (Supplementary Fig. 4). An increase in dominance of the most abundant bacterial species is visible in the ranked SADs of both datasets under sufficiently wet conditions (Supplementary Fig. 4b, c).

**Global patterns of soil bacterial habitat diversity.** Motivated by the general agreement with observations of bacterial richness and the SADs produced by the HM, we used highly resolved global datasets for soil properties, NPP and precipitation as inputs to estimate global patterns of soil bacterial habitat richness (Fig. 4a). Recall that a central element of the model is the link between the number of distinct aqueous habitats per soil volume and soil bacterial richness. Additionally, we considered the sizes of aqueous habitats to yield spatially resolved distributions of the Shannon index of bacterial diversity patterns (Fig. 4b). We note that the modeled soil bacterial diversity follows constraints imposed by local soil-carrying capacity where high bacterial cell numbers are associated with locally high NPP and low cell maintenance requirements. Both diversity indices exhibit spatial patterns with distinct regions of increased diversity associated with climatic transition zones (e.g., the Sahel). This pattern is more pronounced when considering the Shannon index and suggests that soil bacterial community evenness, indicative of how equally habitats are

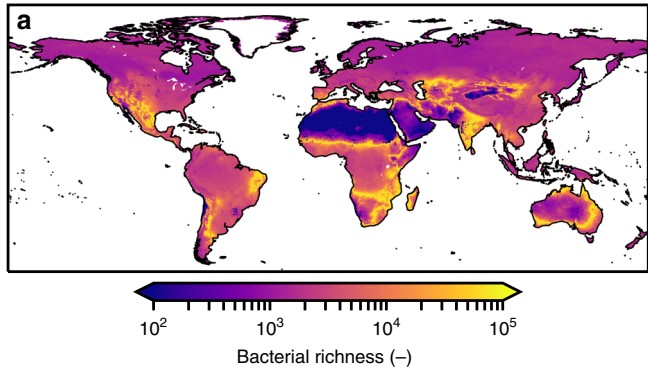

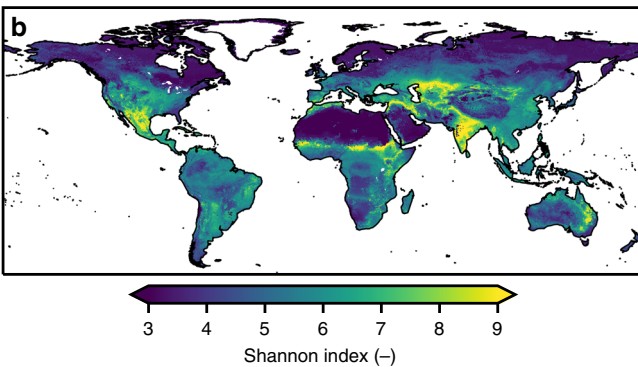

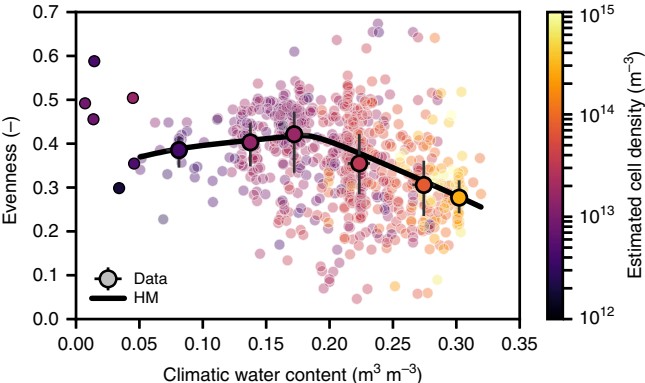

**Fig. 5 Bacterial community evenness decreases with carrying capacity and climatic water contents.** Evenness from two independent studies is shown together with estimated cell density (carrying capacity). Samples were aggregated by latitude, longitude and soil depth (EMP[1], $n = 484$ and DEL[4], $n = 218$). The median and interquartile ranges (colored symbols and bars) are displayed for groups of water contents (bin width: 0.05, number of samples, see Supplementary Table 2). Individual data points are shown for bins containing less than ten samples (small circles) and samples with cell density lower than $10^{12}$ m$^{-3}$ were removed. Evenness predicted by the heuristic model (HM) is calculated using paired values of climatic water content and carrying capacity (evaluated for every sample). Using the joint data of water content and cell density as model input, the HM reproduces the observed tendency of evenness. A locally weighted scatterplot smooth (LOWESS) of modeled evenness is shown for the HM predictions (solid line). Source data are provided as a Source Data file.

**Fig. 4 Modeled global biogeography of soil bacterial diversity.** Global patterns are modeled based on aqueous microhabitats in the top 25 cm considering climate, NPP and soil type. **a** Global map of predicted soil bacterial richness. High values correspond to more heterogeneous soil environments, potentially harboring a larger number of habitats. **b** Global distribution of Shannon index for estimated bacterial diversity. In addition to richness, the Shannon diversity index considers the relative abundance of unique habitats. Higher values of the Shannon index could translate to more even bacterial communities.

partitioned, is sensitive to soil wetness. Such an association is also observed empirically where evenness decreases with increasing climatic water contents (Pearson $r = -0.17$ and $-0.43$ for EMP and DEL, respectively; Supplementary Fig. 5a).

**Disentangling soil bacterial abundance and diversity.** To address the challenge of disentangling bacterial abundance and diversity, we compared bacterial community evenness with climatic water content and carrying capacity (Fig. 5). Evenness decreases gradually with climatic water content and with increasing soil-carrying capacity (Fig. 5, Supplementary Fig. 5b). The results are consistent with the tendency of wetter conditions being associated with an increase in cell densities and was confirmed (with no prior assumptions) using detailed mechanistic modeling (SIM) for small spatial and short temporal scales (Supplementary Fig. 6). In the aqueous-phase fragmentation-based HM, predicted bacterial cell densities are independent of climatic water contents. This could result in unrealistic values relative to empirical observations. We therefore used pairs of values for carrying capacity and climatic water contents to constrain the HM for evenness prediction (Fig. 5). Considering the relation between climatic water content and soil-carrying capacity highlights the sensitivity of HM predictions to bacterial cell density as also observed in the mechanistic simulation results of the SIM. The dependency of cell density on climatic water content in the SIM results in a persistent decrease of evenness with increasing water content (Supplementary Fig. 7). When considering paired values of water content and cell densities obtained

from the SIM, the simpler HM captures the simulated trends reasonably well (Supplementary Fig. 7). Although beyond the scope of this study, we observed that pre-processing measurements of relative species abundance may affect diversity metrics such as richness and evenness, which alters the apparent tendencies (Supplementary Fig. 8).

## Discussion

The heuristic nature of the aqueous-phase fragmentation-based model (HM) precludes comparison of bacterial richness and abundance on a per sample basis, as climatic assumptions and associated large-scale variables are not likely to apply at a particular sampling location and time. Nonetheless, the proposed HM captures the salient features of global trends in bacterial richness related to climate, biome and soil type. Our estimate of soil bacterial cell density represents an upper bound on soil bacterial abundance (carrying capacity) and shows general agreement with measurements of soil bacterial biomass carbon[18]. It tracks the temperature dependency of reaction rates[30] and provides an independent measure of maximal cell density that is sensitive to climate and organic carbon input by vegetation. Bacterial diversity increases towards lower values of climatic water contents (i.e., with increased aridity[15]), as long as soil bacterial life is not limited by low organic carbon input. Assuming a constant soil bacterial carrying capacity, we can attribute much of the variations in bacterial richness to the microscale behavior of soil hydration conditions (Fig. 3). Surprisingly, the trends of bacterial richness for both surveys EMP[1] and DEL[4] were very similar despite their different objectives and processing protocols of the genetic information, namely the use of amplicon sequence variants in EMP and operational taxonomic units in DEL (Fig. 3a, b). We note that the values of bacterial richness in the DEL dataset saturate towards lower values of climatic soil hydration (Fig. 3b). This is likely due to the truncation of species richness used in that study which focuses on the most abundant soil bacteria[4]. These, highly abundant species, might be

the last to disappear under reduced carrying capacity and therefore do not show a decline towards dry conditions. The data available at low climatic water contents are sparse and do not provide support for the predicted steep decline of bacterial diversity as soil becomes dry that was previously reported with increased aridity at large scales[15]. However, a significant decrease in bacterial richness was also observed in a recent statistical meta-analysis for climatic scales[31] and could be confirmed using the SIM (Fig. 3b). Additionally, it has been reported that bacterial diversity declines sharply with moisture in dry soils of Antarctica[23] and decreases with soil relative humidity along transects of the Atacama desert[32]. Microcosm experiments revealed an increase in richness with moisture that peaks at intermediate water contents that promote rare bacterial species[33]. Similarly, bacterial richness was highest at intermediate climatic water contents where isolated aqueous habitats are numerous and sufficiently well supplied by diffusion to realize the soil-carrying capacity (Fig. 3). This observation is supported by the mechanistic simulation results of the SIM, which explicitly considers the dynamics and spatial structure of the bacterial community (Fig. 3). The generality of the aqueous-phase fragmentation-based approach permits comparison of systems with different dimensionality and can account for the shift of maximal richness towards higher water contents when comparing the HM with the two-dimensional simulation of bacterial life on hydrated surfaces by the SIM (Supplementary Fig. 2).

Increasing the organic carbon input and thus soil bacterial abundance seems to support higher diversity of soil microorganisms[15]. This is in line with the observation of decreasing bacterial richness with soil depth (Fig. 3a) that is often attributed to diminishing carbon inputs with depth (Fig. 2b). However, considering the various interacting factors at play, the general picture might be more complicated. An increase in soil-carrying capacity may not necessarily translate to increased bacterial diversity as evidenced by declining community evenness (Fig. 5, Supplementary Fig. 5). This could be due to dominance of a few species that may cluster near nutrient hotspots[34], or loss of oligotrophic species that would be outcompeted in well-connected and dense communities[9]. We observe sensitivity of bacterial evenness to climatic water contents (Fig. 5), also in relation to the soil-carrying capacity (Supplementary Fig. 5). However, care should be taken regarding the interpretation of bacterial richness and evenness, since biases introduced by data processing and sampling could depend on the shape of the underlying SAD (Supplementary Fig. 8). Mechanistic models, such as HM and SIM, are valuable tools to investigate such dependencies as illustrated by considering only the most abundant species (Fig. 3b) or increasing sampling effort and removing species present at low abundance (Supplementary Fig. 8a, b, respectively). Nonetheless, an inherent tradeoff between availability of nutrients and protection by spatial isolation appears to play an important role in the establishment and maintenance of high soil bacterial diversity[17,31,34]. In other words, the relation between bacterial abundance and diversity is only positive when the aqueous phase is fragmented and spatial isolation suppresses the dominance of few species. As aqueous microhabitats become connected following soil rewetting by rainfall or irrigation, competition and other trophic interactions between bacterial cells are likely to reduce soil bacterial diversity (Fig. 3a, b) by reducing the communities evenness (Fig. 5). Many other factors such as pH[1,2,14,19], nutrient composition[5], carbon sources distribution[6,34], stoichiometric constraints[14,18] and metabolic dependencies[35] shape soil bacterial abundance and diversity and could contribute to the discrepancy between our HM and empirical observations. Our study suggests that some of those factors might be associated with climatic hydration conditions. Interestingly, we find that soil samples exhibiting high bacterial diversity at intermediate climatic water contents coincide with near neutral pH values. In contrast, samples at low and high climatic water contents show high (basic) and low (acidic) pH tendencies, respectively (Fig. 3b). This is supported by studies that relate soil pH with differences in soil water balance at climatological timescales[36]. We consequently expect soil pH to result from differences between precipitation and evapotranspiration as described by climatic water contents (Supplementary Fig. 9). Teasing apart such confounding associations requires detailed statistical analysis and experimental validation, which are best conducted in dedicated studies.

Using a single parameter set, largely based on standard percolation theory combined with data on soil properties, our HM predicts SADs that closely resemble empirical observations (Supplementary Figs. 3, 4). Nevertheless, the increased aqueous-phase connectedness in climatically wet soils may also promote interactions that are suppressed under spatial isolation of dry conditions[23]. Processes that support bacterial species coexistence across small distances are not captured by the present model and would result in persistent underestimation of bacterial diversity (unless provisions are introduced as done for very large aqueous habitats—see Supplementary Figs. 2, 3). Another inherent limitation of the analyses presented here is the focus on soil bacteria ignoring the interplay with other soil microorganisms that comprise Earth's microbiome[20]. For example, fungi could play an important role in modifying soil bacterial habitats[2] and are currently only considered in the partitioning of microbial carbon.

The framework presented in this study captures the salient spatial trends in soil bacterial diversity at climatic timescales and provides insights into effects of habitat fragmentation on the prevalence of bacterial interactions in natural soil. This is particularly important for the interpretation of species co-occurrence and interspecific interactions[35]. Such interactions between different species become possible only for conditions supported by the soil aqueous-phase connectedness[23]. This promotes diversity by enabling macroscopic coexistence[5,7,24] in soil bacterial communities competing for space and a common resource.

A unique aspect of the HM is the ability to bridge scales from soil pores to biomes where information at both scales is preserved. Further investigations are required to test some of the model implications at different scales. For example, elucidating the dependency of cell microscale distribution on soil type and hydration conditions could provide insights into the processes shaping bacterial interactions in soil. Additionally taking into account factors affecting the partitioning of carbon at the ecosystem scale could refine model estimates of bacterial abundance beyond potential carrying capacity. Nonetheless, modeling climate and soil-specific bacterial diversity offers a useful reference for comparing effects of climatic shifts (e.g. in temperature, precipitation) or land use change (e.g. in intensity of agricultural management or restoration to natural ecosystems) on soil bacterial communities that could guide future exploration of the soil bacterial micro- and macro geography.

## Methods
In the following, we provide a detailed overview of the methods used in the study and list key assumptions. Although the HM uses a yearly timescale for climatic averaging, the framework could be applied to finer and more resolved datasets. The global predictions of soil bacterial diversity were based on a $0.1° \times 0.1°$ grid to harmonize raster layers. For a description of data sources, see Supplementary Table 1. Variables added to the datasets of point measurements are taken at the native, highest spatial resolution of the respective property. Where necessary and not explicitly stated, missing values were imputed using the mean value of the corresponding variable.

**Soil bacterial carrying capacity derived from NPP.** The flux of carbon into the soil is taken from the MODIS NPP dataset[37]. We have used mean annual values (2000–2015). Missing values (e.g. desert) were imputed with values obtained from the Miami model[38] using parameters fitted to the nonmissing values of MODIS

NPP. Only an average fraction ($\epsilon = 0.24$) of the total NPP entering the soil column is available for bacterial respiration[28,29]. The vertical distribution of microbial carbon in the soil column follows the distribution of plant roots[18]. This allowed us to impose the depth $z$ at which most of the carbon is released by integrating over the sampled interval d$z$ and calculating the fraction of NPP available for bacteria at a particular depth (NPP$_{b,z}$ = $\epsilon \frac{\text{NPP}}{d_{soil}} F_z = \epsilon \frac{\text{NPP}}{d_{soil}} \int f(z) \mathrm{d}z$). The factor $F_z$ denotes the fraction of carbon available at a particular depth and is described by $f(z)$ for the entire depth of the soil profile considered ($d_{soil} = 1$ m). Assuming no net growth of the bacterial community so that only energy requirements for maintenance metabolism are satisfied permits computation of maximal bacterial cell density $\rho_{cell}$ (m$^{-3}$). This soil-carrying capacity supported by the input flux of carbon is calculated using Eq. (1).

$$\rho_{cell}(z, T) = \frac{\text{NPP}_{b,z}}{f_T m M_c}. \tag{1}$$

Using a constant mass of carbon per cell $M_c$ and by fitting maintenance rate $m$, we calculated the bacterial cell density $\rho_{cell}$. Temperature dependency was implemented as a factor $f_T$ based on the Schoolfield model[30] using mean annual temperature (MAT) from the WorldClim dataset[39].

**Soil bacterial abundance dataset**. Xu et al. (XU)[18] compiled a dataset for the abundance of soil carbon associated with microbial biomass. This was used here as a reference for bacterial abundance for a range of geographical locations. We considered the relation between the soils carbon to nitrogen (C:N) ratio and the proportion of bacterial biomass to total microbial biomass[40]. Total microbial biomass carbon contains mainly fungal and bacterial carbon ($C_{mic} \approx C_F + C_B$). A piece wise linear function was used to describe the ratio of fungal to bacterial carbon ($R_{FB} = C_F / C_B$) with varying C:N ratio of the soil organic matter. This ratio was taken as a constant below C:N = 18.4 ($R_{FB} = 5$, see ref. [28]) and increases with a slope of 0.5 above said value[40]. From $R_{FB}$ the relative proportion of bacterial biomass $f_B$ was calculated ($f_B = 1 / (R_{FB} + 1)$). A carbon content per cell[41] of $M_c = 8.6 \times 10^{-14}$ gC was used in all conversions of soil bacterial biomass and for the estimation of soil-carrying capacity. To determine the decay of carbon input in the soil profile ($f_z$) we first averaged the bacterial biomass per soil depth. Averaging was necessary to avoid putting more weight on more frequently sampled depths. Values were integrated from the soil surface to the maximum depth of 2 m. This cumulated bacterial biomass was normalized by its total sum to obtain the cumulative fraction of biomass with soil depth. For parameter estimation, we fit the cumulative lognormal distribution to the cumulative fraction of bacterial biomass yielding $\mu = 0.18$ and $\sigma = 1.00$ for parametrization of $F_z$. We chose a lognormal distribution as it gave a better fit to the vertical distribution of measured bacterial biomass than the previously used exponential model (Supplementary Fig. 1). The global maintenance rate was subsequently estimated by fitting Eq. (1) for the soil-carrying capacity to measurements of soil biomass carbon[18] using inputs of local NPP$_{b,z}$ and MAT. The optimization yielded a maintenance rate of $m = 1.5$ gC gC$_{cell}^{-1}$ y$^{-1}$.

**Soil bacterial diversity datasets**. Two datasets of bacterial species/phylotype abundances based on 16S rRNA sequencing were employed in this study. Data from the Earth Microbiome Project as published by Thompson et al. (EMP)[1] and data collected by Delgado-Baquerizo et al. (DEL)[4] were used to estimate bacterial diversity. Diversity was calculated on the data "as provided" using the procedures outlined below. Except some samples in the EMP dataset had to be removed due to misclassification or unsuitable conditions. The following procedure was applied to filter the EMP data based on metadata: Samples labeled as "Soil(non-saline)" were selected if the environmental material was either "soil" or "bulk soil". We then removed samples containing the features "oil contaminated soil" or "extreme high temperature habitat". Tables of sampled abundances of phylotypes were then used as published (90 bp qc filtered and rarified to 5000 for EMP ($n = 2871$) and the top 511 phylotypes after taxonomic assignment for DEL ($n = 237$)). Variables relevant to soil and climate were added according to reported geographical coordinates and soil depth resulting in 484 and 218 sites for EMP and DEL, respectively. The mass of soil is taken from the extraction protocol used in the studies. For DEL, 0.25 g of soil and for EMP an average of 0.175 g were chosen.

**Estimating soil-specific "climatic" water content**. A metric for the average hydration conditions relies on estimation of a representative value of water content based on rainfall patterns. We use a simplified approach where the periods in which soil drains or dries following a rain event are calculated. We apply a threshold to the precipitation time series to remove small wetting events that immediately evaporate and estimate the time in between rain events. The average duration between events is the characteristic dry down for given geographical locations. During this time, water mass is lost at a constant rate determined by (mean daily) potential evapotranspiration (PET), resulting in an exponential reduction of average water content within the considered soil profile ($d_{soil} = 1$ m). We assume for simplicity that a daily temporal resolution is compatible with the cessation of internal drainage of most soils. Hence, climatic soil water content does not exceed field capacity (a stable water content after internal drainage becomes negligible). For simplicity, we define the volumetric field capacity $\theta_{FC}$ ($V_{water}/V_{soil}$ in m$^3$ m$^{-3}$) as half of the porosity $\theta_s$ ($V_{void}/V_{soil}$ in m$^3$ m$^{-3}$). The latter is obtained

using an empirical (pedo-transfer) function[42] that relates commonly measured soil properties (sand-, silt-, clay- contents and bulk density[43]) to soil porosity. The MSWEP[44] precipitation records of 37 years (1979–2016) are used to derive average rainfall quantities per wetting−drying cycle. The spatial resolution of the precipitation data is roughly 11 km at the equator and the temporal resolution is given at a sub-daily (3 hourly) timescale. The data are down sampled to daily resolution as the dynamics of soil wetting and drying relevant for the bacterial habitat are expected to be within this timescale. Further, the precipitation time series is subjected to a threshold taken from estimates of PET[45] based on temperature and radiation[39] to identify wetting events. The run lengths between wetting events are measured and averaged across wetting cycles. The key result of the analysis is the mean time interval between rainfall events $\tau$ (an ensemble average) for every location. This quantity combined with daily PET (m d$^{-1}$) were used to deduce the climatic water contents $\theta_\tau$ ($V_{water}/V_{soil}$ in m$^3$ m$^{-3}$) according to Eq. (2).

$$\theta_\tau = \theta_{FC} e^{-\alpha<\tau>} \ with \ \alpha = \frac{\text{PET}}{d_{soil}\theta_{FC}}. \tag{2}$$

The significance of $\theta_\tau$ is that it combines rainfall patterns, PET, and soil properties over climatic timescales and provides a measure of the average hydration conditions experienced by soil bacteria in a particular geographical location (Supplementary Fig. 9).

**Estimation of aqueous habitat size distribution**. We estimated the size distribution of distinct aqueous habitats based on soil properties and hydration conditions (e.g., climatic water content). Soil water content was treated as the aqueous-phase occupancy probability $p$ (the probability of finding a water filled pore or roughness element) that, in turn, enabled the application of standard percolation theory to represent the characteristics of aqueous bacterial habitats (sizes and numbers). We considered the soil as a three-dimensional lattice (two-dimensional (2D) for comparison with the SIM) with a critical occupancy probability and universal exponents that determine the number of (aqueous) patches and their sizes[46]. The critical percolation threshold $p_c$ was multiplied by the soil void fraction (or saturated water content $\theta_s$) to account for soil porosity[47]. The critical water content is thus defined by Eq. (3) and could be expressed as critical saturation $S_c$ (4) to remove the dependency on $\theta_s$.

$$\theta_c = \theta_s p_c, \tag{3}$$

$$S_c = \frac{\theta_c}{\theta_s} = p_c. \tag{4}$$

The size distribution of aqueous patches $n_s(p)$ was assumed to follow general proportionalities of percolation theory (5–7)[46]:

$$n_s(p) \sim s^{-\tau} e^{-\frac{s}{s_\xi}}, \tag{5}$$

$$s_\xi \sim |p_c - p|^{-\frac{1}{\sigma}}, \tag{6}$$

$$P^\infty \sim (p - p_c)^\beta. \tag{7}$$

With the patch size $s$ (number of sites/pores) for $s \gg 1$, Fisher exponent $\tau \approx 2.18$ (2D: $\tau = 187/91$), cutoff exponent $\sigma \approx 0.45$ (2D: $\sigma = 36/91$) and cutoff size $s_\xi$[46]. $P^\infty$ is the fraction of the domain occupied by a spanning (algebraically infinite) patch with exponent $\beta \approx 0.41$ (2D: $\beta = 5/36$). The patch sizes follow a power law distribution at $p = p_c$. Away from this critical point when the cutoff size $s_\xi$ is exceeded, patches shrink with decreasing water content ($p < p_c$) or merge and grow when approaching saturation ($p > p_c$) as patches of size $s > s_\xi$ become exponentially scarce. Although the prediction is strictly valid only for $p$ close to $p_c$, we assume such relations to hold for the range of conditions considered. The occupancy probability $p$ was thus substituted with climatic water content $\theta_\tau$ and $p_c$ with a critical water content $\theta_c \approx 0.15$ that correspond to a simple cubic lattice with porosity $\theta_s \approx 0.5$ (triangular lattice in 2D; $\theta_c \approx 0.25$).

To account for different soil types, a characteristic length scale $\delta$ is estimated based on the geometric mean diameter of soil particles[48]. This length scale is used for normalization of the aqueous patch size distribution in the range of water contents and patch sizes relevant for bacterial life. The soil type length scale $\delta$ and the system size $L$ were considered (soil domain or sample size); here we used the mass of soil sampled $m_{soil}$ and bulk-density $\rho_{soil}$ specific to soil type (8). The total number of candidate sites $N_0$ in the sampled soil was then determined from simple geometry considering the dimensionality $d = 2$ or 3 (9).

$$L = \left(\frac{m_{soil}}{\rho_{soil}}\right)^{\frac{1}{d}}, \tag{8}$$

$$N_0 = \frac{L^d}{\delta^d} \tag{9}$$

We approximated the behavior of the percolation transition using a bounded logistic curve that provides a smooth function $\hat{P}^\infty$

$$\hat{P}^\infty = \frac{\theta}{1 + e^{-k(\theta - \theta_c)}}, \tag{10}$$

where $k$ describes the "sharpness" of the transition ($k = 16$ for all calculations). The total size of aqueous clusters or potential habitats $N_s$ was normalized as follows:

$$N_s^0(\theta, N_0) = \frac{\theta - \hat{P}^\infty}{\sum_1^{N_0} s\, n_s(\theta)}, \tag{11}$$

$$N_s(\theta, s) = N_s^0(\theta, N_0) s\, n_s(\theta). \tag{12}$$

Thus requiring, by pre-factor $N_s^0$, that the total volume of aqueous patches conserves the volume of soil water at a given state of hydration. For practical reasons, subsequent calculations of aqueous patches proceed by removing the largest patch after normalization (this large patch biases the counting of habitats in a sample).

**Calculation of bacterial species diversity.** The distribution of aqueous patches derived from percolation theory and their properties defined the degree of spatial isolation and restricted the number of potential habitats. Both aspects were expected to alter the bacterial diversity patterns observed in natural soils. The estimated aqueous patch sizes and their prevalence defined the distribution of bacterial habitats. Together with carrying capacity we can estimate the number of cells within a single (habitat) size class $s$ (13).

$$N_{cell,s} = \rho_{cell}\, s\, \delta^d \tag{13}$$

Aqueous patches with cell counts below a prescribed threshold (or limit of detection, $N_{cell} < 4000$ for comparisons with empirical data) were removed from the total number of potential habitats $N_s$. Conceptually this can be interpreted as the discrete nature of bacterial cells that limits counts to integers greater than one. Empirically, there exists a lower limit of detection and a minimal number of cells from a single species ($\gg 1$) is needed to contribute to the measurement of bacterial richness. Initially, we assumed that only a single species occupies a patch by outcompeting possible coinhabitants. Hereby, the modeled species abundance distribution (SAD) follows the distribution of aqueous habitats with abundances bounded by carrying capacity within a defined volume of soil. Subsequently, we introduced the possibility of multiple species occupying large aqueous patches (in proportion to their size and dimension; $N_{sp} \sim s^{1/d}$, $d = 2$ or 3) to correct for model bias of over predicting the dominant species. The exponent ($1/d$) suggests that the number of species per habitat grows with the average distance between any two points selected randomly within a single habitat of size $s$. The limit of detection was not used for the comparison of SADs as the total number of habitats was truncated to the number of observed species.

Bacterial diversity was calculated in the general form[49] for all SADs (modeled and data):

$$^qD = \left(\sum_{i=1}^{SR} p_i^q\right)^{1/(1-q)} \tag{14}$$

With relative species abundance $p_i$ and species richness SR. For $q = 0$ the equation corresponds to the weighted harmonic mean and equals the actual number of types (SR). The equation is not defined for $q = 1$ where the limiting form is described by the well-known Shannon index $H$ (15) and evenness $E_{1,0}$ is calculated as defined by Eq. (16)[49].

$$\lim_{q \to 1} {}^qD = {}^1D = \exp(H) = \exp\left(-\sum_{i=1}^{SR} p_i \ln(p_i)\right) \tag{15}$$

$$E_{1,0} = \frac{{}^1D}{{}^0D}. \tag{16}$$

**Spatially explicit individual-based model (SIM).** An individual-based approach was previously developed to model growth of diverse bacterial species on heterogeneous soil surfaces[7,25] and was adopted for the current study. The spatial domain was represented by a hexagonal grid with periodic boundary conditions (length $L = 1$ mm; area of a grid cell $A_{hex} = 100\ \mu m^2$; and porosity $\theta_s = 0.49$). Grid cells consisted of water holding elements with volumes drawn from a random uniform distribution (unif) that have a maximal size equal to the spacing of the grid ($dx = 1.1 \times 10^{-5}$ m). Thereby the modeled domain represents a slab of the soil pore space with a defined volume ($V_{soil} = L^2\, dx$). The bulk water content is prescribed to the domain as a control parameter and spatially distributed relative to the sizes of grid elements while conserving the total volume of water ($V_{water} = \Sigma\, V_{water,x,y}$). Based on the local volume of water, an average water film thickness $h$ was calculated ($h_{water,x,y} = V_{water,x,y}/A_{hex}$). The heterogeneity of the water film thickness modified the mass transfer between grid cells by changing the cross-sectional area that contributed to the diffusive flux. Diffusion was solved using the implicit finite differences method with bacterial consumption represented as a sink term. Diffusivity is taken for a small molecule that is readily available for bacterial consumption (e.g. glucose) and does not vary spatially ($D = 6.7 \times 10^{-10}$ m$^2$ s$^{-1}$). The simulation period corresponded to 8 days at a 1-min time step. Initial concentration of nutrients was constant in space and randomly replenished to initial concentration over time to mimic a fluctuating environment. The arrival of nutrient pulses was modeled as a Poisson process with an average rate of one arrival every 4 h. The initial nutrient concentration was set to provide enough carbon to sustain a fixed cell density ($10^{17}$ m$^{-3}$, corresponding to high carrying capacity) and was distributed evenly among nutrient pulses. The mass of nutrients locally available for bacterial consumption depended on the volume of water in a grid cell. All simulated bacteria were represented as elongating cylindrical capsules that consume a common carbon source dissolved in the aqueous phase. The diversity and multiple species $i$ were prescribed in the model by varying Monod parameters (growth rate $\mu_{max,i}$, half saturation constant $K_i$—additionally maintenance rate $m_i := 0.01\ \mu_{max,i}$). Species-specific parameters were randomly selected from uniform distributions of the Monod parameters ($\mu_{max} \sim$ unif($10^{-4}$ h$^{-1}$, 1.14 h$^{-1}$), $K \sim$ unif(6.8 g m$^{-3}$, 680 g m$^{-3}$)). All other parameters were held constant (mass of the cell $m_{cell} = 9.5 \times 10^{-13}$ g, mass at division $m_{div} = 2\ m_{cell}$, yield $Y = 0.5$, cell radius $r_{cell} = 0.5\ \mu m$). A single cell of each species was inoculated randomly on the domain at the beginning of the simulation (species richness SR at $t = 0$, SR$_{t0} = 4096$). Individual cells grew and divided along their axis with a slight asymmetry in mass to avoid complete synchrony ($f_m \sim$ unif(0, 0.05), $m_{cell,1} = f_m m_{div}$ and $m_{cell,2} = (1 - f_m)m_{div}$). All bacterial cells were subject to active and passive motion and could move continuously in the domain. Growth-induced shoving represents the passive motion and was implemented by displacing cells relative to their nearest neighbors (only considering the capsule geometry as $n$-spheres; no forces, e.g. capillary, friction, elastic, electrostatic, etc.). Shoving was not resolved to full relaxation due to the size of the domain, number of cells and the scale of interest (compromise between reduced computational demand and precision of the resulting spatial distributions). However, we implemented a simple rule to prevent local crowding: if the projected area of bacterial cells in a grid cell exceeded the area of the grid cell ($A_{hex}$), bacterial cells were randomly picked and moved to form a second layer (piling cells at the $z$-direction) from which they could "drop" down again once space became available. Bacterial swimming motility was permitted where the aqueous phase was connected and the water film thickness exceeded cell diameter[26]. Cells aligned their motility trajectories along gradients of the nutrient field, whereas their velocity was modified by the water film thickness[26] and nutrient concentration[50]. Additionally, each velocity component ($v_x$, $v_y$) is independently multiplied with a random factor to allow for individual trajectories ($f_v \sim$ unif(0, 2)). Integrating along the projected trajectory of each cell enabled consideration of varying water film thickness and prevented cells with high instantaneous velocity from "jumping" across grid cells. At the end of the simulation, the total number of cells and the number of cells per species were measured. To enable comparison of richness estimates from varying sample sizes (e.g. with observed species richness or simulations with different cell densities), total cell numbers were rarified to 5000 and 1000 counts, to compare with EMP and DEL, respectively. For comparison with the DEL dataset only the top 512 most abundant species were considered. Singletons, i.e. cells that were sampled only once when rarefying, were removed from the counts. The rarefaction procedure was averaged across 15 trials to increase robustness of the diversity estimates. Only community evenness was also estimated without rarefaction and removal of singletons as it affected the apparent community structure (Supplementary Fig. 8).

**Reporting summary.** Further information on research design is available in the Nature Research Reporting Summary linked to this article.

## Data availability
The source data underlying Figs. 2, 3 and 5 are provided as a Source Data file. All generated data (by the SIM) are available from the corresponding author (samuel.bickel@usys.ethz.ch) upon request.

## Code availability
Custom computer code is accessible online (for the HM, https://gitlab.ethz.ch/bickels/microgeo-ncomms, and SIM, https://gitlab.ethz.ch/bickels/microgeo-sim) and is archived in a public repository (https://doi.org/10.5281/zenodo.3558542).

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

## Acknowledgements

We gratefully acknowledge Oskar Hagen and Minsu Kim for helpful comments on early drafts of the manuscript and critical reflection and discussion on the conceptual framework. This work was funded by the European Research Council (ERC) Advanced Grant "SoilLife" (No 320499) and the MicroScapesX (SystemsX.ch) and carried out at ETH Zürich.

## Author contributions

D.O. and S.B. designed research; D.O. and S.B. performed research; S.B. wrote computer code; S.B. performed experiments. D.O. and S.B. carried out the analysis of results and wrote the paper.

## Competing interests

The authors declare no competing interests.
