## [Peer Review File · Nature Communications]

Reviewers' Comments:

Reviewer #1:

Remarks to the Author:

In their manuscript "Soil bacterial diversity shaped by microscale processes across biomes" the authors present a model that – as they claim – allows to calculate the biodiversity in soil from the properties of the local soil and climate. This results in a simulated map of microbial diversity of the (nearly) whole planet. I think the overall approach is very interesting. I have never seen a similar approach before (although I am also not exactly working in the same field).

However, I had unfortunately quite a hard time with the paper. Especially it was unclear to me what in the end the model is and how it works. Moreover, it is hard to understand what is really the predictive power of the model, which makes the whole approach very speculative for me. Also the writing of the paper is often hard to follow. I think the manuscript would strongly profit from clearer more direct language with less jargon, a clearer description of what was actually done and assessment of the quality of the model.

Major points:

I remains unclear to me how the calculation of the micro-habitat distribution and carrying capacity come together to generate the outcomes that are shown. In several cases the parameters remain unclear and contradicting statements about the model are made (more detailed points see below).

In general there are no real statements about the model quality. Like deviation between model and measurements or prediction by alternative models. Without that it is very hard to say if the model has any predictive power at all.

The maintenance rate m is fitted locally to the data as far as I understood. Later on this local m is used to predict species diversity at a certain location. It feels for me the authors use data to generate prediction about exactly this data and thus move in a circle. I think the way to test the predictive power of a model is to make predictions about data that was not used for fitting parameters.

As far as I understand the authors assume that there is one species in each micro-habitat. Therefore if the carrying capacity is changed the total population density of each species changes but the diversity should stay the same, because the relative abundance stays the same. So why is for the model incorporation of NPP, MAT and their effect on carrying capacity necessary when the carrying capacity has not influence on relative abundance?

I think species abundance often follows a power law distribution and also percolation theory delivers power law distributions. Therefore it seems not surprising to me that SAD can show correlation with habitat size distribution (Extended Fig.1), but it does not mean the one causes the other. For example also a scale free interaction networks may lead to a power law shaped SAD.

The writing style is often very complicated. I had to read many sentences multiple times. I feel in many cases there are logical connections between statements missing. Moreover, the model in the main text is explained very vaguely. I had to read the supplement to understand the main text (at least partially, as said above I still do not understand what the whole model does).

Further points:

In Fig. 1 the biodiversity is not influenced by the water content, which is in contradiction to what the authors claim.

Fig. 2: I feel it is very hard to tell from that figures if the model is in line with the data. I feel many functions would do the job and there is no discussion how well the model fits, like correlation or test of alternative models.

Fig.3 Again I do not feel that the model aligns well with the data. The model shows a maximum of species diversity with a water content of 0.2 (by the way what are units?) which I cannot see in the data. Again no metrics are used to show how well the model describes the data or other models are used.

Fig3: I thought the carrying capacity is calculated for every sampling location from MAT and NPP? Shouldn't it be possible to compare every sampled data-point to a calculated form the model and not work with a range of carrying capacities instead?

Fig.4 I don't understand what is plotted here. How is habitat richness defined? How are microhabitats distinguished (how much to they have to differ in size to be different)? In b) there is once written it shows the diversity of the habitats then the diversity of the bacteria?

Line78. What happens if you use a high value for carbon per cell? Why did you choose to lose a low value?

Methods:

Formula [1] to what is m fitted?

You treat the bacterial density as a function of depth? Where do the authors get fz from?

In the supplement the authors state that the source of organic carbon is uniformly distributed. How does this fit together?

Line 252: Shouldn't PET be something like volume per time? Its given as m/d? Whats is the unit of the field capacity?

Formula 2: As far as I understand this formula is derived by assuming a exponential evaporation of water from the ground between the rain (τ). However, shouldn't formula 2 then be an integral over an exponential decay which should have the form $1/\alpha \cdot (1 - \exp(-\alpha \tau))$? Shouldn't the formula the authors provide just give the water content after the time τ ?

Line 241: For simplicity we define the field capacity θ as half of the porosity θ obtained using a pedo-transfer function

Maybe you could shortly explain what that means?

line 271: Why a log-normal model? I feel Fig2b could be fit with many functions?

Line 267-270: I don't understand these sentences.

How were the NPPs split? Randomly? By any criterion?

Where do you get the single MAT value from? How is it chosen? Averaged?

95 percentile of what? What did you rank?

Lin271: Again I have strong issues to understand.
What did you normalize the bacterial biomass?
What is the lower depth of the sampling interval?

Line341: The samples are not taken from single micro-habitats. So why using a detection threshold of the sampling/sequencing for micro-habitats?

How should I understand dsoil? Is there rock below this 1m of soil?

In the supplement there is a bunch of discussion about different lengthscales in the soil. How does this matter for the model?

Supplement line 603: Soil structural properties, such as texture and porosity, provide characteristic length scales important for describing the small scales relevant to soil bacterial diversity. What does it mean, how is it incorporated into the model?

Supplement line 610: Limiting interactions to competition only, enables unambiguous ranking of species by their growth rates with the expectation of a single species dominating each aqueous patch. What does that mean? Growth rate and competition are not necessarily connected.

Supplement line 624: By selecting appropriate maintenance rates, we may obtain an upper bound for cell density that vary among geographic locations (integrating NPP and temperature). I assume that's not a integration in a mathematical sense but means that both are taken into account?

Supplement 627: To relax this assumption, we have considered a prescribed fraction (25%) of the NPP dedicated to soil bacteria.
Isn't the ratio of bacteria and fungi strongly varying eg with the pH of the soil?

Supplement line 634: Additionally, we cannot ignore the dependency of NPP on precipitation and do not attempt to treat derived carrying capacity as an independent variable but rather as a location specific property
I don't understand that sentence

Supplement 654: Soil microbial abundance declines rapidly with depth due to a commensurate decrease in nutrient availability as determined by plant litter and root exudates.
Didn't you say earlier that you assume homogeneous input of carbon across the depth of 1m?

Figure S1: SAD is a curve and D a number, so how can D be proportional to SAD?

How does NPP influence diversity? I see how it influences total abundance but not necessarily diversity. Is it because at low NPP carrying capacity is low and thus some micro-habitats are left empty?

Extended figure 1: What are units of axes? What were parameters of models? Was there any fitting

done? For example what is factor for proportionality between species abundance and size of micro-habitat? How was it obtained?

Reviewer #2:

Remarks to the Author:

Biogeography studies on soil microbial community are developing fast and providing many exciting results. Yet, they remain somehow disappointing because regarding soils characteristics, only chemical characteristics are considered in those studies, such as pH, available nutrients, C, C/N... It is however well established from local scale studies that the abundance, diversity and activity of soil microbial communities are also largely influenced by physical characteristics of their habitat, which are spatially very heterogeneous at the microscale. Indeed, as written by the authors, it was demonstrated that the hydration states of soil aqueous microhabitats, their size distribution and connectedness shape bacterial abundance and diversity.

In this context, the work presented in this manuscript is highly welcome and relevant. Stating that there is only anecdotal evidence of a link between soil carbon and soil biodiversity is exaggerated and not necessary to justify the work developed here (soil bacterial diversity does increase with the organic matter content, as shown by biogeography studies from the local (e.g. Siciliano et al. 2015), the regional (e.g. Maestre et al. 2015, Pasternak et al. 2013, Liu et al. 2014) and the global (Delgado-Baquerizo et al. 2016) scales).

The authors propose to explain the biogeography of soil microorganisms by trophic conditions and by microscale aqueous habitats. For this they combine in each point of continents an estimate of the trophic carrying capacity for microbes (defined as a certain proportion of the NPP), with a model describing local microbial habitat conditions (number of aqueous microhabitats and their size distribution), to predict the abundance and diversity of soil bacterial and compare these predictions with published data on abundance and diversity of soil bacteria. As a conclusion, most of the variations in soil bacteria diversity are ascribed to the microscale hydration conditions.

The approach is very attractive. The work is with no doubt novel and very creative. It is extremely exciting to try to bridge microscale and macro scales microbial biogeography. Yet the manuscript does not convince me, because it goes too far too fast. As a result, it is also very frustrating for the reader. Describing the trophic conditions and microscale physical conditions that bacteria experience in soils of the whole planet with available data worldwide at the targeted resolution ($0.1 \times 0.1^\circ$) requires a number of simplifications. Among them it is stated that the trophic resources for bacteria are 25% of NPP whatever the ecosystem, that field capacity is half of soil total porosity in all soils of the world, considered that only one bacterial "species" stand in a given wet microhabitat, etc. Why not after all, if it is to develop the proof of concept... However, it becomes very difficult to follow the results with the many successive assumptions and no discussion of their limits and consequences step by step and not presentation of intermediate results. The reader cannot appreciate the consequences of the many assumptions.

It is not satisfying that the approach is developed at such a large scale that no validation is possible, while it should be possible developing it locally, considering e.g. 3-5 sites with contrasted soils, well documented for all the variables needed here, the variables being measured, not modelled, before going global. Then for example it would be possible to test the hypotheses considered to be verified on the global dataset (e.g. "a compensating effect of enhanced carrying capacity that allows higher richness despite an increase in habitat connectedness", l. 168). In addition, only complex final variables are presented in figures and there are no "intermediate" results presented e.g. on the number of aqueous microhabitats and their size distribution depending on soil type and moisture conditions for contrasted soil and climate conditions. I am not a soil physicist and cannot evaluate the model presented here, but would have liked intermediate results to be presented and discussed.

I found it difficult along the manuscript to differentiate results, from discussion and to follow precisely the discussion (e.g. section lines 160-183), especially as the literature is treated in a very general way, not accounting if the results cited were obtained from a microcosm model study, or from a plot or region scale.

In conclusion, I find that this work is novel and important. It is important enough to strengthen, validate, discuss it stepwise, e.g. on a number of case studies at the plot scale before going global, and I would suggest to do so in a newly submitted version.

Reviewer #3:

Remarks to the Author:

The manuscript by Bickel and Or describes an interesting theoretical, modeling-based study of soil bacterial biodiversity patterns and provides an interesting perspective in which micro-scaler habitat connectivity and carrying capacity are combined. The work is laudable in that it attempts to link micro-scale patterns to bacterial diversity at the biome level. The authors develop models of bacterial abundance and diversity patterns and examine the accuracy of their models using two large-scale datasets of bacterial microbiome diversity.

In general, I think the work has potential value, but also suffers from some important drawbacks in my opinion. Perhaps most importantly, I found it difficult to decipher out the clear take-home messages that the authors wish to put forth. This is for instance an issue in the abstract: from the abstract it is difficult to determine what was actually done in the study and which conclusions are most important. Also, there are some parts of the text that remain somewhat vague (I try to point out a few below). In addition, the study is somewhat limited in its appreciation for other factors driving bacterial diversity patterns; while this is mentioned, it would be helpful to have the current study put more into the context of these different drivers.

In total, I very much appreciate the authors' quantitative approach to integrating micro-scale patterns of habitat connectivity and bacterial abundance with larger scale patterns of microbial diversity, but I think the manuscript would have to be improved substantially to get its message across effectively to a broad audience.

Below, I have listed a number of issues for the authors' consideration (not in order of importance).

- 1.) L26 and throughout: it may seem a bit nit-picky, but please be careful with the use of the term population. This should refer to a species, not a group of species
- 2.) L27-29: I'm not exactly sure what you mean here – do you refer to the shape of the species-area curve (steep at first, then flat, then rising again)?
- 3.) L32: You refer here to key soil factors, but only consider a couple – I suggest being clearer and referring to the specific factors you examined.
- 4.) L33: This is a bit vague – it would first have to be determined that bacterial diversity and abundance are “entangled” (in what way?) before we would have a need to disentangle them. I think it would be easier and more straightforward to simply state the question as “what determines the relative diversity of soil bacterial communities”?
- 5.) L39: Again – what exactly is the “challenge of ecosystem functional diversity”? Also, you claim that it is important to be able to predict soil-borne microbial diversity – can you make a stronger case for the need to be able to do this?
- 6.) L56: It might be handy to state what you start with this simplification.
- 7.) L63-65: It might be useful to explain here (and more in the final section) that you do not examine

other properties like pH, disturbance, etc. i.e. it is not that you seek to ignore other factors, but that you seek to examine specifically habitat connectivity and density.

8.) L90-L92: could this also have to do with the loss of many aerobic populations?

9.) L103-105 (and in general): Is it possible to tease apart these confounding associations?

10.) L137: This seems like a strange expression given the fact that the Shannon index is calculated from richness and evenness.

11.) L145: Why does this refer to "bacterial" biomes? As far as I can tell, this refers to biomes in general.

12.) L153: It seems to me that land use would be of great importance – how does that fit into your scheme?

13.) L187-189: Might however be worth mentioning that volatile compounds can also be important here & these would only be effective in less saturated soils.

14.) L193-195: Again, how does the current study relate to the importance of these factors in driving patterns of soil-borne bacterial diversity?

15.) L226: This assumption is obviously not constant – how would this affect the model?

16.) Fig3b: Why does this relationship break down at low climatic water content for the DEL database? This deserves some discussion.

Herewith also a few very minor things:

1.) L13: hyphenate "Biome-specific"

2.) L44: delete "a"

3.) L51: Delete "In"

4.) L62: hyphenate "long-term"

5.) L67: insert "our" before "model"

Reviewers' comments:**Reviewer #1 (Remarks to the Author):**

“In their manuscript “Soil bacterial diversity shaped by microscale processes across biomes” the authors present a model that – as they claim – allows to calculate the biodiversity in soil from the properties of the local soil and climate. This results in a simulated map of microbial diversity of the (nearly) whole planet. I think the overall approach is very interesting. I have never seen a similar approach before (although I am also not exactly working in the same field).”

We thank the reviewer for the general appreciation of our approach. To the best of our knowledge it is the first process based model that attempts to link soil and climatic properties to soil bacterial diversity across scales. The simplified approach allows estimation of bacterial diversity (at the global scale) for locations where soil and climate properties are available.

“However, I had unfortunately quite a hard time with the paper. Especially it was unclear to me what in the end the model is and how it works. Moreover, it is hard to understand what is really the predictive power of the model, which makes the whole approach very speculative for me. Also the writing of the paper is often hard to follow. I think the manuscript would strongly profit from clearer more direct language with less jargon, a clearer description of what was actually done and assessment of the quality of the model.”

In the revised manuscript we attempted to clarify the points of concerns raised by the reviewer. We attempted to streamline the presentation by simplifying sentences and minimizing the use of jargon where possible. Supplementary text was removed and necessary information was incorporated into the main text for clarity of the modeling approach (introduction and methods). Additionally, the model introduction (L52-74) was revised and a new illustrative figure was included (Fig. 1). At the core of the model framework is the estimation of numbers and sizes of bacterial habitats that depend on soil type and climate. Together with an estimate of carrying capacity, we can model soil bacterial diversity that is sensitive to hydration conditions. We supplemented this aqueous phase fragmentation-based heuristic model (HM) with a spatially-explicit individual-based mechanistic model (SIM) for comparison. The SIM consider cell-cell trophic interactions, growth, motility and other life functions of individual cells of different bacterial species based on physiological growth parameters. The SIM makes no *a-priori* assumptions regarding the relations between numbers of aqueous habitats and bacterial diversity and the resulting diversity in the simulation domain is an emergent property of the biophysical interactions. Using the simulations of the SIM we evaluated predictions of the HM and find good agreement between the two approaches and empirical observation (see revised Fig. 3).

“Major points:

I remains unclear to me how the calculation of the micro-habitat distribution and carrying capacity come together to generate the outcomes that are shown. In several cases the parameters remain unclear and contradicting statements about the model are made (more detailed points see below).”

We clarified these points in the revised manuscript. The calculations of micro-habitat occupancy were done using eq. 12 that describes the size and numbers of aqueous patches (serving as potential unique habitats) as a function of water content. We use eq. 13 to estimate the number of bacterial cells per habitat size class depending on local cell density (soil carrying capacity). Combining this information with the assumption on how many species share a single micro-habitat (in the simplest case, we allow for one species to dominate the habitat) we obtain estimates of species abundance distribution (SAD). The resulting SAD which is sensitive to water contents and carrying capacity, provide the basis for calculating other metrics of diversity. Most parameters are taken from standard percolation theory or average soil properties and have not been fitted to data. A single parameter set (for simple cubic and triangular lattices in three and two dimensions, respectively) was used for both datasets of bacterial diversity with soil properties specific to each dataset. Parameters pertaining to percolation theory only are reported in the methods section (L351-371) while average values specific to the used datasets are reported under results (L114-117) in the revised version of the manuscript. The combination of carrying capacity and micro-habitat distribution, i.e. the calculation of the SAD, is described in the Methods section (see “Calculation of bacterial species abundance distribution and species diversity”).

“In general there are no real statements about the model quality. Like deviation between model and measurements or prediction by alternative models. Without that it is very hard to say if the model has any predictive power at all.”

The reviewer is correct in the observation, yet, to the best of our knowledge, there are no similar models that include soil and climatic properties and permit direct comparison with our HM. Most existing ecological models that describe SADs rely on empirical fitting parameters that are difficult to interpret with respect to the physical structure of habitats (e.g. neutral models¹) or require many implicit assumptions on species properties (e.g. trait-based, idiosyncratic models²). Furthermore, results of either approach would be indistinguishable with respect to the SAD². The novelty in this study is to use the HM as mechanistic framework that provide a basis for explaining changes in the SAD based on carrying capacity and climatic water content. Nevertheless, to address this and other comments we have used a mechanistic model that simulates microbial life on hydrated soil surfaces³⁻⁶ to evaluate the HM predictions independently (at least for the small scale). The resulting diversity and abundance trends by the HM were confirmed using the more detailed

and computationally demanding SIM results (albeit with focus on shorter temporal and smaller spatial scales). While we cannot rule out additional factors, our findings suggest that the physical aqueous phase configuration plays an important role in shaping bacterial cell-cell interactions, ranges of migration, spatial confinement and more that ultimately determine soil bacterial diversity. We could, in principal, provide goodness-of-fit metrics to compare our HM with other (statistical) models. However, since we are not fitting our model to the diversity data it is not meaningful to do so. Additionally, we use inferred and remotely sensed soil and climatic properties that are subject to large uncertainty and we do not expect to reproduce exactly local estimates of diversity.

“The maintenance rate m is fitted locally to the data as far as I understood. Later on this local m is used to predict species diversity at a certain location. It feels for me the authors use data to generate prediction about exactly this data and thus move in a circle. I think the way to test the predictive power of a model is to make predictions about data that was not used for fitting parameters.”

We think that this is a misunderstanding; the maintenance rate m is estimated globally using a dataset of microbial biomass carbon⁷ that is independent of the bacterial 16S diversity datasets^{8,9}. While m is estimated globally, local information on mean annual temperature (MAT) is used to adjust the rate m and account for different geographic locations. Apart from m , two additional parameters that affect variations in carrying capacity with soil depth were fitted to the dataset of soil microbial biomass carbon⁷. All other parameters were held constant. Additionally, we note that local carbon inputs and soil type are also important variables in determining the resulting soil bacterial abundance.

“As far as I understand the authors assume that there is one species in each micro-habitat. Therefore if the carrying capacity is changed the total population density of each species changes but the diversity should stay the same, because the relative abundance stays the same. So why is for the model incorporation of NPP, MAT and their effect on carrying capacity necessary when the carrying capacity has not influence on relative abundance?”

As we discussed in the original version, the assumption of a single species dominating a microhabitat was made as a base case to evaluate the HM and determine potential SAD. Later in the study, we relaxed this assumption based on observations that deviate from model predictions especially under wet conditions (SI Fig. S3). The soil carrying capacity in the HM is used to constrain the number and sizes of potential habitats (i.e. isolated aqueous patches that can contain a certain number of cells). It is correct that in the process of scaling the relative SAD by cell density should not change diversity. Measurements of bacterial diversity, however, are subject to a limit of detection (cell counts of a single species). By applying a cutoff in cell density, small habitats do not contribute to measures of diversity. This relation with carrying capacity is sensitive to the shape of the SAD and causes

abundance and diversity to be “entangled”. In our HM we use a cutoff in number of cells to remove habitats that are unlikely to be occupied or contain a too low number of cells to be detected. The SAD is hereby sensitive to carrying capacity and thus NPP and MAT.

“I think species abundance often follows a power law distribution and also percolation theory delivers power law distributions. Therefore it seems not surprising to me that SAD can show correlation with habitat size distribution (Extended Fig.1), but it does not mean the one causes the other. For example also a scale free interaction networks may lead to a power law shaped SAD.”

For clarification, we note that percolation theory delivers a power law at the critical point only. Away from this the SAD is represented by an exponential cutoff sensitive to climatic water contents and with magnitudes in richness that are sensitive to soil type (total potential number of habitats). Models of SAD that generate similar power laws with an exponential cutoff include neutral and idiosyncratic models². Both assume the existence of a meta-population (prior distribution) from which species can immigrate from. In soils, we argue, that the aqueous phase is fragmented and thus disconnected. To establish a power law distribution within a single aqueous patch it would be necessary to fulfill requirements of meta-population based models (i.e. im-/migration between patches is possible). While other mechanisms could lead to a power law-like distribution, the process based HM model generates SADs that closely follow empirical observations. Additionally, results from the SIM (mechanistic model) that make no *a-priori* assumptions about the nature and consequences of aqueous phase fragmentation result in similar SAD. Evidence suggest that the strong physical constraints in the complex soil environment shape the interaction between microbial cells by limiting aqueous diffusion and direct cell-cell contacts. This could be an underlying cause of a scale free interaction network (by physically confining interactions to power law distributed aqueous patches).

“The writing style is often very complicated. I had to read many sentences multiple times. I feel in many cases there are logical connections between statements missing. Moreover, the model in the main text is explained very vaguely. I had to read the supplement to understand the main text (at least partially, as said above I still do not understand what the whole model does).”

We thank the reviewer for the feedback on the writing style and for highlighting the need of providing logical connections and model descriptions. The entire manuscript text was revised accordingly. We split long and complicated sentences and added explanatory paragraphs where applicable.

Further points:

“In Fig. 1 the biodiversity is not influenced by the water content, which is in contradiction to what the authors claim.”

Figure 1 has been completely reworked and now shows the influence of aqueous habitats on bacterial diversity.

“Fig. 2: I feel it is very hard to tell from that figures if the model is in line with the data. I feel many functions would do the job and there is no discussion how well the model fits, like correlation or test of alternative models.”

We adapted the calculations for figure 2 and provided a more accessible presentation of results by displaying the effects of temperature on the modeled and observed cell densities (Fig. 2a solid lines and kernel density estimates). We are not aware of any comparable process based models that would allow estimation of carrying capacity (upper bound on cell density) and could be included. Regarding the distribution of bacterial biomass with soil depth, we have used the log-normal model as it provided a better fit than the previously reported exponential model (SI fig. S1). We further compared the dependency of carrying capacity on water contents with simulations using the SIM (SI fig. S6). Note that in both models (HM, SIM) the relation between water contents and cell density is not explicitly prescribed.

“Fig.3 Again I do not feel that the model aligns well with the data. The model shows a maximum of species diversity with a water content of 0.2 (by the way what are units?) which I cannot see in the data. Again no metrics are used to show how well the model describes the data or other models are used.”

We note that the samples from which the data are derived came from different locations and soil depths hence, all we can hope to capture are trends (no single line would fit the data). In Fig. 3a of the revised manuscript, we labeled the sample depth and applied the HM to the mean carrying capacity (solid red line) as well as to the low values associated with large depths (dashed black line). The HM model estimates bracket the observations and capture (qualitatively) the general trends with climatic water content. To compare modeled trends in diversity (Fig. 3 - red lines), empirical estimates have been grouped by soil depth (Fig. 3a) and water contents (Fig. 3 a and b). Water contents are volumetric (V_{water}/V_{soil} in m^{-3}/m^{-3}) and missing units have been added to figure legends and the methods section. The lower diversity in deeper soil layers is expected due to the reduction in carrying capacity and is captured by the HM (Fig. 3a dashed line). The simple HM provides mean trends of bacterial diversity comparable with two independent datasets of bacterial diversity by using a single parameter set and without attempting to fit the data. We are not aware of comparable models and thus report simulation results using the SIM (Fig. 3 – blue squares). The SIM

provides qualitatively similar results with a discrepancy in water contents corresponding to maximal richness that could be ascribed to the differences in dimensionality between the HM and SIM (SI Fig. S2).

Fig. 3. Observed and predicted variations in soil bacterial diversity with climatic water content (m^{-3}/m^{-3}). **a** and **b**, Empirical estimates of bacterial richness from two studies are binned by climatic water contents (bin width: 0.05) and the median and interquartile range are reported (colored circles and bars, respectively). Exact number of samples per group are listed in table S2. The solid red lines correspond to predictions by the fragmentation-based heuristic model (HM) for median carrying capacity specific to each dataset. Blue squares and bars (mean \pm SD, $n = 10$) depict simulated bacterial richness using the spatially-explicit individual-based model (SIM) for different water contents. **a**, Bacterial richness from the Earth Microbiome Project (Thompson *et al.* - EMP)⁸ was reported for different soil depths and thus grouped accordingly (5, 15, 30 and 60 cm). The dashed line represents a model scenario with reduced carrying capacity by considering only the subsoil (20-100 cm). **b**, Soil bacterial richness from a recent study (Delgado-Baquerizo *et al.* - DEL)⁹. The colors indicate estimates of soil pH¹⁰, which has been shown to be affected by climate¹¹. For comparison with the DEL dataset, only the top 512 species were considered in the SIM.

“Fig3: I thought the carrying capacity is calculated for every sampling location from MAT and NPP? Shouldn’t it be possible to compare every sampled data-point to a calculated form the model and not work with a range of carrying capacities instead?”

This goes along the expectation that the HM model could describe soil bacterial diversity values for specific soil samples – such specific prediction is beyond the capability of any available model at preset. While NPP and MAT are local properties, they are also derived from large scale observations and might not necessarily reflect conditions at small scales. Based on the heuristic assumptions that underlie such a simple model, we do not expect cell densities to be at carrying capacity at every location in space and time where bacterial diversity was sampled. Soil carrying capacity is used to take into account the carbon input and temperature of biomes by scaling bacterial diversity relative to a lower cutoff (limit of detection). Our focus lied on the role of hydration conditions in affecting bacterial diversity and we found the median carrying capacity to be representative for average trends (Fig. 3).

“Fig.4 I don’t understand what is plotted here. How is habitat richness defined? How are microhabitats distinguished (how much do they have to differ in size to be different)? In b) there is once written it shows the diversity of the habitats then the diversity of the bacteria?”

In the revised manuscript we clarify the assumed equivalency between habitat richness (numbers and sizes of unique aqueous habitats) and bacterial richness (see results section L143-144). This is a core assumption that links the number of habitats and the number of bacterial species in a soil volume. For the simplest case where we assume one species dominating each habitat, we could substitute habitat richness with bacterial richness (otherwise certain provisions need to be made to link these two with a modified calculation). The size of a habitat depends on the soil type (textural characteristic length δ , L356-357) and on the mean water content in the soil. Both define the smallest unit length for distinguishing habitats of varying sizes. The ambiguity in Fig. 4b was removed by writing “[...]estimated bacterial habitat diversity.”.

“Line78. What happens if you use a high value for carbon per cell? Why did you choose to lose a low value?”

Generally speaking, any changes in the amount of carbon per cell would shift the soil carrying capacity values but these could be partially compensated by the (re-)fitted maintenance rate m . In any case, evidence suggest that the values are within a narrow range and are unlikely to change the order of magnitude of cell densities (due to the limited range of cell sizes¹²). We updated the previously used value by changing to a literature reference that contains estimates more relevant for soil environments (from $M_c = 2 \times 10^{-14} \text{ g C}^{13}$ to $8.6 \times 10^{-14} \text{ g C}^{14}$).

Methods:

“Formula [1] to what is m fitted?”

The procedure of obtaining m is described in L262-268 and L286-289 in the methods section. We use local MAT and soil depth specific NPP to fit m to measured estimates of bacterial biomass carbon.

“You treat the bacterial density as a function of depth? Where do the authors get f_z from? In the supplement the authors state that the source of organic carbon is uniformly distributed. How does this fit together?”

We thank the reviewer for the opportunity to clarify this important point. The “uniform distribution of carbon” referred to the very small scales (size of a single aqueous habitat) and not to the distribution along the soil profile. The distribution of bacterial biomass carbon with soil depth is parametrized using a log-normal distribution (providing f_z). We have re-written the methods section and removed the supplementary text.

“Line 252: Shouldn’t PET be something like volume per time? Its given as m/d ? Whats is the unit of the field capacity?”

Terrestrial fluxes are often expressed in hydrology in units of length per time (which in effect are units of water volume per area of land per time). Field capacity, the amount of water held in a soil against gravity, is expressed as volumetric water contents ($V_{\text{water}}/V_{\text{soil}}$ in m^3/m^3).

“Formula 2: As far as I understand this formula is derived by assuming an exponential evaporation of water from the ground between the rain (τ). However, shouldn't formula 2 then be an integral over an exponential decay which should have the form $1/\alpha \cdot (1 - \exp(-\alpha \tau))$? Shouldn't the formula the authors provide just give the water content after the time τ ?”

Yes, the formula gives the water content after the characteristic time τ (an ensemble average). We adapted the methods to state this more explicitly (L325-332).

“Line 241: For simplicity we define the field capacity θ as half of the porosity θ obtained using a pedo-transfer function. Maybe you could shortly explain what that means?”

A sentence to explain how porosity was obtained was added (L314-316): “The latter [porosity] is obtained using an empirical (pedo-transfer) function that relates commonly measured soil properties (sand-, silt-, clay- contents and bulk density) to soil porosity.” We assume climatic water contents cannot exceed field capacity as the time scale considered would allow the soil to drain internally (L315-317).

“line 271: Why a log-normal model? I feel Fig2b could be fit with many functions?”

In principle, the reviewer is correct, however using previously reported exponential model^{7,15} resulted in a poor fit compared to the log-normal (SI Fig S1). Often top soils are dry and contain fewer roots than deeper in the soil (hence affecting soil carbon distribution). The misfit of an exponential model was particularly evident for the top soil (10 cm) where most of samples (abundance and diversity) were taken. We agree that other bounded functions with “heavy” tails could be appropriate. Nonetheless, we opted for the log-normal as it is the most parsimonious (multiplicative random process) and provides tractable central tendencies (mean of the log normal corresponds to the median of the untransformed data).

“Line 267-270: I don't understand these sentences. How were the NPPs split? Randomly? By any criterion? Where do you get the single MAT value from? How is it chosen? Averaged? 95 percentile of what? What did you rank?”

The language was sloppy and we meant to explain the partitioning of NPP and the other parameters. We have revised the entire section to clarify the procedure and assumption in simpler language (see Materials and methods in the revised manuscript).

“Lin271: Again I have strong issues to understand. What did you normalize the bacterial biomass? What is the lower depth of the sampling interval?”

Soil samples are usually taken from a depth interval (e.g. 5-10 cm) but are reported as an average value for the average depth (midpoint of the interval). Our previous calculation relied on using the lower depth of this interval to accumulate values of bacterial biomass carbon. This was not necessary for the calculation and could be simplified using the mean values directly. The entire section was rewritten and the simplified calculations do not rely on the lower depth interval for normalization (see Materials and methods in the revised manuscript).

“Line341: The samples are not taken from single micro-habitats. So why using a detection threshold of the sampling/sequencing for micro-habitats?”

We use a threshold (number of cells) to detect the occupancy of a micro-habitat size class. In other words if micro-habitats do not contain a sufficient number of cells they are unlikely to contribute to measures of diversity. The need for using a detection threshold has been clarified in the methods section (L386-390): “Aqueous patches with cell counts below a prescribed threshold (or limit of detection, $N_{cell} < 3600$ for comparisons with empirical data) were removed from the total number of potential habitats N_s . Conceptually this can be interpreted as the discrete nature of bacterial cells that limits counts to integer values. Empirically, there exists a lower limit of detection and a minimal number of cells from a single species ($\gg 1$) is needed to contribute to the measurement of bacterial richness.”

“How should I understand dsoil? Is there rock below this 1m of soil?”

For purposes of harmonizing data and predictions across biomes and scales we define a prescribed cutoff depth for soil hydrological and biological processes. Most of the plant root system and associated soil microbial processes are contained within the top 1 m^{7,15}. Although, soil depth might be restricted to 1 m at certain locations (e.g. higher altitudes) we use this value to designate the soil volume where we would expect soil hydration conditions to be climatically constant and where most of the carbon input flux would be distributed (e.g. more than 90% of bacterial biomass is contained within 1 m, SI Fig. S1).

“In the supplement there is a bunch of discussion about different lengthscales in the soil. How does this matter for the model?”

The discussion was motivated by the need to link the various natural scales for soil hydrological and biological processes. We agree with the reviewer regarding the limited value added by discussing the cascade of scales for this study, hence we removed this supplementary text from the revised manuscript.

“Supplement line 603: Soil structural properties, such as texture and porosity, provide characteristic length scales important for describing the small scales relevant to soil bacterial diversity. What does it mean, how is it incorporated into the model?”

This is an important point, the idea is that coarse textured soils (sand) would have fewer grains and lower surface area relative to fine textured soils (this difference is encapsulated in the textural characteristic length δ). The impacts of these soil specific differences affect the estimated number of potential habitats (eq. 9) for the same water content across different soil texture by adjusting the smallest habitat size.

“Supplement line 610: Limiting interactions to competition only, enables unambiguous ranking of species by their growth rates with the expectation of a single species dominating each aqueous patch. What does that mean? Growth rate and competition are not necessarily connected.”

The reviewer is correct in that trophic interactions and growth rates are not inherently linked, yet in small systems such a single aqueous patch (under our model assumptions) the outcome of the competition could be determined by the physiological traits of the species in the patch with limited resources. For example, when considering affinity¹⁶ (growth rate/half saturation constant) that measures the competitive ability at low substrate concentrations. The questionable line was removed together with the supplementary text.

“Supplement line 624: By selecting appropriate maintenance rates, we may obtain an upper bound for cell density that vary among geographic locations (integrating NPP and temperature). I assume that's not a integration in a mathematical sense but means that both are taken into account?”

The supplementary text was removed and the ambiguity was resolved in the Materials and methods section. We distinguish the integration of NPP over a depth interval (to calculate the amount of carbon released at a particular depth) from the inclusion of temperature in the calculations of carrying capacity.

“Supplement 627: To relax this assumption, we have considered a prescribed fraction (25%) of the NPP dedicated to soil bacteria. Isn't the ratio of bacteria and fungi strongly varying eg with the pH of the soil?”

We thank the reviewer for pointing out an ambiguity in our representation of the partitioning of carbon input (net primary productivity - NPP) to bacterial respiration and the attribution of soil microbial biomass carbon to fungal and bacterial biomass carbon. We revised the manuscript to clearly distinguish the mean respiration fraction of bacteria, for calculations of carrying capacity, and the attribution of microbial carbon to fungi and bacteria, for conversions of measurements of microbial carbon⁷ to bacterial cell densities. A recent study found an average proportion (24% of NPP) that is respired by bacteria from mechanistic modeling¹⁷ in agreement with empirical observations¹⁸. Although, a slight increase of respired NPP would be expected¹⁷, an average value was considered representative for a wide range of NPP (L78-81, L259-262). The attribution of soil microbial biomass carbon to fungi and bacteria is more sensitive environmental conditions^{17,19}. The strong relation of fungal to bacterial biomass ratio with environmental conditions¹⁹ was considered in the revised version of the manuscript for conversions of microbial biomass carbon to bacterial cell densities (L277-282). Briefly, we used the soil C:N ratio to vary the attribution of microbial biomass carbon to fungal and bacterial biomass carbon^{17,19}.

“Supplement line 634: Additionally, we cannot ignore the dependency of NPP on precipitation and do not attempt to treat derived carrying capacity as an independent variable but rather as a location specific property I don't understand that sentence”

Net primary productivity (NPP) is an ecosystem level property that considers regional vegetation patterns and is determined by climatic conditions (among other factors). We used precipitation data to estimate climatic water contents and encountered an association with carrying capacity (calculated based on NPP and MAT). Thus, we cannot ignore the possible relations of carrying capacity with precipitation (or climatic water contents) in calculations of bacterial diversity and do not treat them as independent properties (but use both to describe conditions at a geographic location). Briefly, the original statement corresponds to the association of NPP with climatic water contents that shows an increasing tendency under the conditions considered. The relation between carrying capacity and climatic water content is reported in the results section (L159-169) of the revised manuscript. The corresponding, original sentence and the supplementary text have been removed.

“Supplement 654: Soil microbial abundance declines rapidly with depth due to a commensurate decrease in nutrient availability as determined by plant litter and root exudates. Didnt you say earlier that you assume homogeneous input of carbon across the depth of 1m?”

There is a misunderstanding of the scales where we assumed homogeneous carbon input – the statement of homogeneous input applies at the microscale, however, a microhabitat at a depth of 0.2 m would have more carbon than a microhabitat at a depth of 1 m. At a scale of microhabitats carbon inputs are homogeneous and at a soil profile scale, carbon inputs decline rapidly with soil depth and was parametrized using a log-normal distribution.

“Figure S1: SAD is a curve and D a number, so how can D be proportional to SAD?”

We thank the reviewer for finding this mistake. It should read: $D \sim f(SAD)$. However, we have removed the supplementary text and illustrative figure S1.

“How does NPP influence diversity? I see how it influences total abundance but not necessarily diversity. Is it because at low NPP carrying capacity is low and thus some micro-habitats are left empty?”

Yes, at low cell densities small habitats fall below the limit of detection (are unlikely to contain cells) and do not contribute to measures of diversity.

“Extended figure 1: What are units of axes? What were parameters of models? Was there any fitting done? For example what is factor for proportionality between species abundance and size of micro-habitat? How was it obtained?”

A single parameter set was used as described in the Materials and methods of the revised manuscript. Three parameters have been fitted to data of bacterial biomass carbon: two parameters for the log-normal distribution that describe the decay in cell density with depth, and the maintenance rate m used for conversion of carbon input flux (NPP) to cell density at carrying capacity. The factor for proportionality describing the number of species per microhabitat is prescribed considering the dimensionality.

Reviewer #2 (Remarks to the Author):

“Biogeography studies on soil microbial community are developing fast and providing many exciting results. Yet, they remain somehow disappointing because regarding soils characteristics, only chemical characteristics are considered in those studies, such as pH, available nutrients, C, C/N... It is however well established from local scale studies that the abundance, diversity and activity of soil microbial communities are also largely influenced by physical characteristics of their habitat, which are spatially very heterogeneous at the microscale. Indeed, as written by the authors, it was demonstrated that the hydration states of soil aqueous microhabitats, their size distribution and connectedness shape bacterial abundance and diversity. In this context, the work presented in this manuscript is highly welcome and relevant.”

We thank the reviewer for the encouraging comment. The motivation of our work was to develop the ability of bringing scales from soil grains to biomes while considering the soils heterogeneity at small scales.

“Stating that there is only anecdotal evidence of a link between soil carbon and soil biodiversity is exaggerated and not necessary to justify the work developed here (soil bacterial diversity does increase with the organic matter content, as shown by biogeography studies from the local (e.g. Siciliano et al. 2015), the regional (e. g. Maestre et al.2015, Pasternak et al. 2013, Liu et al. 2014) and the global (Delgado-Baquerizo et al. 2016) scales).”

We thank the reviewer for pointing out literature references and agree that the wording was inadequate. We adapted the paragraphs in question (L26-30).

“The authors propose to explain the biogeography of soil microorganisms by trophic conditions and by microscale aqueous habitats. For this they combine in each point of continents an estimate of the trophic carrying capacity for microbes (defined as a certain proportion of the NPP), with a model describing local microbial habitat conditions (number of aqueous microhabitats and their size distribution), to predict the abundance and diversity of soil bacterial and compare these predictions with published data on abundance and diversity of soil bacteria. As a conclusion, most of the variations in soil bacteria diversity are ascribed to the microscale hydration conditions. The approach is very attractive. The work is with no doubt novel and very creative. It is extremely exciting to try to bridge microscale and macro scales microbial biogeography. Yet the manuscript does not convince me, because it goes too far too fast. As a result, it is also very frustrating for the reader.”

While we are delighted by the reviewer's recognition of our works novelty, we disagree that the study goes “too far too fast” in the context of spatial scales. The goal of this study is to establish general trends at large spatial scales across a wide range of environments. Many properties cannot be separated (e.g. carrying capacity and hydration conditions, SI fig. S6) and lead to the necessity of incorporating additional formulations (e.g. parametrizing the decay of biomass with soil depth, SI fig. S1) that incorporate the environmental context. We thus emphasized that while we have used data

from a wide range of conditions (global datasets), the model captures trends that are also predicted by a spatially-explicit individual based model (SIM). In summary, the HM is aimed at presenting general regional trends based on mechanistic understanding of microscale conditions, and is not intended to provide specific predictions for a single sample (this would be beyond the capability of most mechanistic and statistical soil bacterial life models available at present; L173-175). We focus on few, in our opinion, important aspects (hydration conditions and biome specific carrying capacity) that are sufficient for understanding the main conclusions (aqueous micro-habitat fragmentation mediates soil bacterial diversity) and capture general trends of soil bacterial abundance and diversity.

“Describing the trophic conditions and microscale physical conditions that bacteria experience in soils of the whole planet with available data worldwide at the targeted resolution (0.1x0.1°) requires a number of simplifications. Among them it is stated that the trophic resources for bacteria are 25% of NPP whatever the ecosystem, that field capacity is half of soil total porosity in all soils of the world, considered that only one bacterial “species” stand in a given wet microhabitat, etc. Why not after all, if it is to develop the proof of concept... However, it becomes very difficult to follow the results with the many successive assumptions and no discussion of their limits and consequences step by step and not presentation of intermediate results. The reader cannot appreciate the consequences of the many assumptions.”

The reviewer touches upon an important point – the necessary simplifications and assumptions for building the simplest model that captures the phenomenon under study (i.e. linking soil bacterial diversity and abundance to mechanistic processes and variables). Ability to link soil bacterial diversity to a “universal variable” that reflects climate, soil type and hydration is the core novelty of the HM propose here. We have revised the explanations and justification of the many assumptions and their context in terms of scale. We make use of certain observations such as that the water content at field capacity across many soil types is about half the value of saturated water content (porosity); we also invoke (for simplicity) the fact that in any aqueous patch single or multiple successful species would emerge as dominating that connected landscape (based on local conditions and physiological traits) – all of these are “heuristic” assumptions that enable us to construct a general heuristic model. We revised the main text to state assumptions more explicitly and to provide brief discussion of their consequences where applicable.

“It is not satisfying that the approach is developed at such a large scale that no validation is possible, while it should be possible developing it locally, considering e.g. 3-5 sites with contrasted soils, well documented for all the variables needed here, the variables being measured, not modelled, before going global. Then for example it would be possible to test the hypotheses considered to be verified

on the global dataset (e.g. “a compensating effect of enhanced carrying capacity that allows higher richness despite an increase in habitat connectedness”, l.168).“

We agree with the reviewer that such an incremental approach would be fine and could bridge the large scale gap between the sample and global scales. The main limitation to pursuing such suggestions are lack of data from a sufficiently wide range of environmental conditions. Our heuristic model (HM) offers a conceptual framework for linking scales where regional scale variables (climate, soil, carbon) become boundary conditions for the microscale where bacterial diversity emerges and is expressed. The goal of this study was not to test every possible hypothesis implied by the HM but to evaluate large scale climatic trends. Those general tendencies could be confirmed with a mechanistic model and are supported by available data. Many factors might affect bacterial diversity and we focus only on few that are necessary to provide model estimates comparable with the observed variation. Both, models and empirical observations, point towards the important role of soil hydration conditions in structuring bacterial habitats. The heuristic nature of our framework and its limitations are explicitly discussed in the revised manuscript (L173-175).

“In addition, only complex final variables are presented in figures and there are no “intermediate” results presented e.g. on the number of aqueous microhabitats and their size distribution depending on soil type and moisture conditions for contrasted soil and climate conditions. I am not a soil physicist and cannot evaluate the model presented here, but would have liked intermediate results to be presented and discussed.”

We chose to not present intermediate results as they are not crucial to explain the main conclusions of the paper. All model elements used to arrive to the main conclusions are contained in the manuscript and we cannot include many more figures due to length restrictions. However, we are willing to provide specific intermediate results upon request.

“I found it difficult along the manuscript to differentiate results, from discussion and to follow precisely the discussion (e.g. section lines 160-183), especially as the literature is treated in a very general way, not accounting if the results cited were obtained from a microcosm model study, or from a plot or region scale.”

We separated the results and discussion sections in the revised manuscript. Additionally, there were changes to the literature cited to better reflect the scales being discussed.

“In conclusion, I find that this work is novel and important. It is important enough to strengthen, validate, discuss it stepwise, e.g. on a number of case studies at the plot scale before going global, and I would suggest to do so in a newly submitted version.”

We thank the reviewer for the encouraging comments. We went great lengths in validating our HM using mechanistic predictions by the SIM. However, we find the global scale appropriate to apply the model framework as we expect relations to establish at large spatial and climatic temporal scales. We hope our revised manuscript can provide additional confidence in the approach.

Reviewer #3 (Remarks to the Author):

“The manuscript by Bickel and Or describes an interesting theoretical, modeling-based study of soil bacterial biodiversity patterns and provides an interesting perspective in which micro-scaler habitat connectivity and carrying capacity are combined. The work is laudable in that it attempts to link micro-scale patterns to bacterial diversity at the biome level. The authors develop models of bacterial abundance and diversity patterns and examine the accuracy of their models using two large-scale datasets of bacterial microbiome diversity.”

We thank the reviewer for complimenting the core of our study and acknowledging the efforts in linking micro-scale patterns of soil aqueous habitats to biome characteristics. We went one step further in this revised version by not only evaluating model predictions of our HM with empirical observations but also providing mechanistic simulations using the SIM.

“In general, I think the work has potential value, but also suffers from some important drawbacks in my opinion. Perhaps most importantly, I found it difficult to decipher out the clear take-home messages that the authors wish to put forth. This is for instance an issue in the abstract: from the abstract it is difficult to determine what was actually done in the study and which conclusions are most important.”

The abstract was rewritten entirely with focus on emphasizing the main message. Additionally we adapted the discussion to more clearly highlight findings and main conclusions.

“Also, there are some parts of the text that remain somewhat vague (I try to point out a few below). In addition, the study is somewhat limited in its appreciation for other factors driving bacterial diversity patterns; while this is mentioned, it would be helpful to have the current study put more into the context of these different drivers.”

We emphasized the role of other factors in the revised introduction (L33-34) and discuss them in context of our process-based understanding (L197-204, L214-219).

“In total, I very much appreciate the authors’ quantitative approach to integrating micro-scale patterns of habitat connectivity and bacterial abundance with larger scale patterns of microbial diversity, but I think the manuscript would have to be improved substantially to get its message across effectively to a broad audience.”

Motivated by the reviewer’s appreciation, we revised the manuscript text substantially and amended model validation by comparing to simulation of the SIM.

“Below, I have listed a number of issue for the authors’ consideration (not in order of importance). 1.) L26 and throughout: it may seem a bit nit-picky, but please be careful with the use of the term population. This should refer to a species, not a group of species”

We thank the reviewer for pointing out this inaccuracy in terminology and have adapted the wording accordingly where applicable.

“2.) L27-29: I’m not exactly sure what you mean here – do you refer to the shape of the species-area curve (steep at first, then flat, then rising again)?”

The referenced section has been removed from the revised manuscript text.

“3.) L32: You refer here to key soil factors, but only consider a couple – I suggest being clearer and referring to the specific factors you examined.”

The key soil factors under consideration have been explicitly stated in the introduction (L30-32) and throughout the method section.

“4.) L33: This is a bit vague – it would first have to be determined that bacterial diversity and abundance are “entangled” (in what way?) before we would have a need to disentangle them. I think it would be easier and more straightforward to simply state the question as “what determines the relative diversity of soil bacterial communities?”

We explained how bacterial diversity and abundance are possibly “entangled” in the introduction (L27-30) and provide a dedicated section in the results (L157-171). Further we found that species abundance and diversity (specifically richness and evenness) are not independent (Fig. 5). This affects measures of richness by making abundant species more detectable and was confirmed using the SIM (SI fig S8). Thus, the processing of diversity data and the measurements themselves are sensitive to the shape of the SAD.

“5.) L39: Again – what exactly is the “challenge of ecosystem functional diversity”? Also, you claim that it is important to be able to predict soil-borne microbial diversity – can you make a stronger case for the need to be able to do this?”

We revised the questionable phrase and provide a stronger case for the need of studying soil bacterial diversity in the context of ecosystem processes (L36-38).

“6.) L56: It might be handy to state what you start with this simplification.”

We added a statement on the consequences of this simplification in the introduction (L56-61).

“7.) L63-65: It might be useful to explain here (and more in the final section) that you do not examine other properties like pH, disturbance, etc. i.e. it is not that you seek to ignore other factors, but that you seek to examine specifically habitat connectivity and density.”

A more explicit statement on the factors considered in this study is provided (L66-70) and we discuss potential limitations (L214-217 and L226-233).

“8.) L90-L92: could this also have to do with the loss of many aerobic populations?”

We cannot rule out a loss of aerobic populations. However, the soils considered are likely to be aerated since samples are mostly taken from top soils (upper 10 cm) and the soils were not fully saturated. Furthermore, we could expect that facultative anaerobes and many aerobes would experience a competitive advantage at higher water contents. This might affect richness in unpredictable ways. Additionally, even if richness would drop due to loss of aerobic species we would still observe a decrease in evenness that suggests that dominance of species is enhanced in wetter environments (SI fig S5). Lastly, if aerobic populations were to be outcompeted it would implicitly require that they share their habitat with better adapted, anaerobic populations. This is in line with the notion of reduced soil bacterial diversity with increased habitat connectedness and would more likely occur in large habitats under wet conditions.

“9.) L103-105 (and in general): Is it possible to tease apart these confounding associations?”

It is difficult and in some cases impossible to tease apart such confounding associations, especially without process based models. Nonetheless, it should be possible to identify hierarchies in variables that affect soil bacterial diversity (e.g. soil pH is proxy of the soils buffering capacity which results from the water balance at climatological timescales¹¹) using mechanistic modelling (and experimental manipulation) to disentangle (and validate) confounding associations.

“10.) L137: This seems like a strange expression given the fact that the Shannon index is calculated from richness and evenness.”

The definition of evenness is given as the diversity of order $q=1$ (exponential of Shannon index) divided by the diversity of order $q=0$ (richness) as described in the Methods section (eq. 16).

“11.) L145: Why does this refer to “bacterial” biomes? As far as I can tell, this refers to biomes in general.”

In general, there is substantial overlap between the traditional definitions (based on temperature and precipitation); but for soils with different water holding capacities our classification would differ. However, the section on bacterial biomes was removed from the manuscript as it did not contribute substantially to the main message.

“12.) L153: It seems to me that land use would be of great importance – how does that fit into your scheme?”

Land use could be readily incorporated in the current model if it changes the effect of climatic variables (e.g. modified hydration conditions due to irrigation) and soil properties (e.g. bulk density due to compaction). Further, many agricultural practices change soil structure (e.g. tillage) and vegetation properties (e.g. crop rotations) that also affect the input of carbon in the soil profile. We discuss few aspects throughout the final section of the revised manuscript.

“13.) L187-189: Might however be worth mentioning that volatile compounds can also be important here & these would only be effective in less saturated soils.”

Considering transport limitations as a function of soil hydration would indeed be interesting as a way to expand the current model and we thank the reviewer for mentioning the subject. However, we are unaware of potential implications for bacterial diversity that would not be mediated by habitat connectedness. We focus on the role of aqueous microhabitat fragmentation and did not discuss implications for gaseous transport in unsaturated soils as it is beyond the scope of this study.

“14.) L193-195: Again, how does the current study relate to the importance of these factors in driving patterns of soil-borne bacterial diversity?”

The roles of other factors are discussed in context of soil bacterial diversity and the findings of our study in the revised manuscript (L214-224).

“15.) L226: This assumption is obviously not constant – how would this affect the model?”

It is not obvious that this assumption is not, at least on average, constant at the scales considered. The value of carbon content per cell is used to convert total bacterial biomass carbon to cell counts. The value is not likely to change the order of magnitude of estimated carrying capacity. It would shift the total number of cells to higher values if a low value would be considered but does not alter the shape of the relation with NPP, MAT or climatic water contents. Thus, it does not affect the central conclusion that aqueous micro-habitat fragmentation affects soil bacterial diversity.

“16.) Fig3b: Why does this relationship break down at low climatic water content for the DEL database? This deserves some discussion.”

We thank the reviewer for this interesting question. From our understanding, the data used in the DEL study considers only the most abundant species. Those are less sensitive to reduced carbon

input (and hydration conditions). Using our SIM we emulated the data processing by truncating the ranked SAD to the top 512 most abundant species and could confirm the invariance of bacterial richness at low to intermediate water contents (Fig. 3b). This is further discussed in the context of disentangling abundance and diversity (L200-206).

“Herewith also a few very minor things:

- 1.) L13: hyphenate “Biome-specific”*
- 2.) L44: delete “a”*
- 3.) L51: Delete “In”*
- 4.) L62: hyphenate “long-term”*
- 5.) L67: insert “our” before “model”*”

The text was substantially revised and we tried to incorporate the suggested changes where applicable.

References

1. Volkov, I., Banavar, J. R., Hubbell, S. P. & Maritan, A. Neutral theory and relative species abundance in ecology. *Nature* **424**, 1035 (2003).
2. Pueyo, S., He, F. & Zillio, T. The maximum entropy formalism and the idiosyncratic theory of biodiversity. *Ecol Lett* **10**, 1017–1028 (2007).
3. Wang, G. & Or, D. Hydration dynamics promote bacterial coexistence on rough surfaces. *The ISME journal* **7**, 395–404 (2013).
4. Kim, M. & Or, D. Individual-Based Model of Microbial Life on Hydrated Rough Soil Surfaces. *PLOS ONE* **11**, e0147394 (2016).
5. Tecon, R., Ebrahimi, A., Kleyer, H., Levi, S. E. & Or, D. Cell-to-cell bacterial interactions promoted by drier conditions on soil surfaces. *PNAS* **115**, 9791–9796 (2018).
6. Borer, B., Tecon, R. & Or, D. Spatial organization of bacterial populations in response to oxygen and carbon counter-gradients in pore networks. *Nature Communications* **9**, 769 (2018).
7. Xu, X., Thornton, P. E. & Post, W. M. A global analysis of soil microbial biomass carbon, nitrogen and phosphorus in terrestrial ecosystems. *Global Ecology and Biogeography* **22**, 737–749 (2013).
8. Thompson, L. R. *et al.* A communal catalogue reveals Earth’s multiscale microbial diversity. *Nature* **551**, 457–463 (2017).
9. Delgado-Baquerizo, M. *et al.* A global atlas of the dominant bacteria found in soil. *Science* **359**, 320–325 (2018).
10. Hengl, T. *et al.* SoilGrids250m: Global gridded soil information based on machine learning. *PloS one* **12**, e0169748 (2017).
11. Slessarev, E. W. *et al.* Water balance creates a threshold in soil pH at the global scale. *Nature* **540**, 567–569 (2016).
12. Kempes, C. P., Wang, L., Amend, J. P., Doyle, J. & Hoehler, T. Evolutionary tradeoffs in cellular composition across diverse bacteria. *ISME J* **10**, 2145–2157 (2016).
13. Simon, M. & Azam, F. Protein content and protein synthesis rates of planktonic marine bacteria. *Marine Ecology Progress Series* **51**, 201–213 (1989).
14. Whitman, W. B., Coleman, D. C. & Wiebe, W. J. Prokaryotes: the unseen majority. *Proceedings of the National Academy of Sciences* **95**, 6578–6583 (1998).
15. Jackson, R. B. *et al.* A global analysis of root distributions for terrestrial biomes. *Oecologia* **108**, 389–411 (1996).
16. Button, D. K. Affinity of organisms for substrate. *Limnology and Oceanography* **31**, 453–456 (1986).
17. Fatichi, S., Manzoni, S., Or, D. & Paschalis, A. A mechanistic model of microbially mediated soil biogeochemical processes - a reality check. *Global Biogeochemical Cycles* (2019). doi:10.1029/2018GB006077
18. Fierer, N., Strickland, M. S., Liptzin, D., Bradford, M. A. & Cleveland, C. C. Global patterns in belowground communities. *Ecology Letters* **12**, 1238–1249 (2009).
19. Waring, B. G., Averill, C. & Hawkes, C. V. Differences in fungal and bacterial physiology alter soil carbon and nitrogen cycling: insights from meta-analysis and theoretical models. *Ecology Letters* **16**, 887–894 (2013).

Reviewers' Comments:

Reviewer #1:

Remarks to the Author:

Overall the manuscript improved a lot, the language is much clearer, the modeling process can be better understood, simulation and data can be better compared and the new SIM model adds valuable information.

I have few minor comments left:

Two times when the model fails a new (usually fitted) parameter is introduced to save it. This makes it difficult to compare the results throughout the paper, since the model basically changes.

Especially the question arises if Fig. 3 would look different allowing more than one species per habitat.

Also in Fig. 5 it remains unclear how using the empirical input parameters changes the outcome and how this model change would affect the other results of the paper.

Fig2: Colors indicate 2-.5,50,97.5 percentile: That was quite confusing for me . The lines in the main plot correspond to specific temperatures (the percentiles) but the kernel densities correspond to temperature ranges. So I wonder how the two sets of curves are comparable and if they should have the same colors?

Fig. 5: For me its hard to see a decrease of evenness in the data. I think if the authors believe there is a decrease they should give a slope of a line fit and its error or similar.

Some suggestions about the figures (that can be ignored):

Maybe use the word 'model' or similar in the title to manage expectations a little

Fig1: maybe show more similar looking species in right panel of soil comic.

Fig2: maybe color datapoints according to temperature range

Fig3: Line 74 estimates of soil pH: how was it estimated? Measured?

Fig5: maybe label SIM and HM symbols according to cell density

Reviewer #3:

Remarks to the Author:

I found the manuscript to be much improved and much more readable and accessible than the original. I think the authors do a good job showing that this general approach is relevant to predicting patterns of microbial diversity. They also do a better job of discussing the limitations and assumptions of their approach.

I think especially the abstract could still do a better job of zooming in on the question at hand, so as to capture the attention of the reader and clearly guide him/her to what is going to be addressed in the manuscript. For instance, the first sentence is so general as to not be very useful. This could be

stated much more sharply. Perhaps something like "microbial diversity has been shown to vary across terrestrial habitats, with presumptive links to function." Also, it would be helpful to clearly state the question that is being addressed.

Herewith some specific comments as I went back through the manuscript:

- 1.) Introduction: the structure of the introduction would be improved by the use of paragraphs.
- 2.) L21: some studies suggest even higher numbers – might be worth including.
- 3.) L24: I think it would be better to refer to "the rare components of the soil microbiome"
- 4.) L62 (and elsewhere) – I think it is important to make it clear that you refer to terrestrial biomes – also, I think this sentence should be rearranged: thus... Modeled trends of soil bacterial carrying capacity and diversity were compared to empirical observations across different terrestrial biomes"
- 5.) L80-82L I suggest switching the order around of this sentence – we found that varying the range of expected values had little impact on carrying capacity estimates. We therefore used a constant value in our model.
- 6.) L202: I think you should define what you mean by hotspots – hot spots for what? Activity? Interaction? Diversity?
- 7.) L223-224: This could of course also be approached experimentally.
- 8.) L247: On the more positive side, you might also mention restoration efforts (as opposed to only aspects that cause soil degradation).
- 9.) Figure 1: I think the drawing might be improved by also included the soil matrix – it is not clear what the white background depicts. L550; later you say that his assumption does not hold, so I think this has to be toned down here. Also, I think that you can make a better distinction between potential carrying capacity and realized carrying capacity.
- 10.) Figure 3: Empirical data is extremely sparse at the dry end of the spectrum (zero and one point in panels a and respectively) – thus the "real" data not show strong support for the sharp diversity decline under the driest conditions here. I think this deserves mention.

Minor corrections/suggestions:

- L13: change "vary" to "varies"
- L32; change "for" to "toward"
- L37: change "deciphering" to "understanding"
- L54: delete "diverse" - delete "The" from next sentence
- L97: delete; "In the following"
- L106: change "additionally" to "also"
- L109: add commas after "dataset" and "depth"
- L114: change to read, "... data, but rather are..."
- L125: change "shift" to "discrepancy" and change "is" to "can be"
- L131: change "to" to "for each"
- L137: change "thus" to "therefore"
- L149: add "patterns" after "diversity"
- L151: change "exhibits" to "exhibit"
- L152: change "and appear" to read ", with this patterns being"
- L159: insert "with" before "carrying" – and change next sentence to start "These results are..."
- L177: change "has shown" to "shows"
- L191: I suggest using the word "realize" instead of "express"
- L197: change "seem" to "seem"
- L232: change "inhabit" to "comprise"
- L555: change to read "Soil bacterial abundance in relation to net..."
- L592: delete "is"

Reviewers' comments:**Reviewer #1 (Remarks to the Author):**

R1.1: *Overall the manuscript improved a lot, the language is much clearer, the modeling process can be better understood, simulation and data can be better compared and the new SIM model adds valuable information.*

We thank the reviewer for the encouraging comment and we fully agree that the additional mechanistic modeling results helped support the heuristic and simple model and also clarified certain important aspects in the revised manuscript.

I have few minor comments left:

R1.2: *Two times when the model fails a new (usually fitted) parameter is introduced to save it. This makes it difficult to compare the results throughout the paper, since the model basically changes.*

We generally agree with the reviewer, yet keeping in mind the minimalistic nature of the heuristic model and the broad range of conditions explained by this relatively simple approach, the performance of the HM is quite remarkable. We wish to clarify that we did not fit an additional parameter to address model limitations; instead, we explored alternative assumptions regarding species occupancy. The original and simplest “single species per aqueous habitat” assumption holds well for most unsaturated conditions where the soil aqueous phase is fragmented to many small habitats. However, as soil water content increases and the aqueous phase becomes reconnected, habitats may grow substantially and are able to accommodate occupancy of multiple bacterial species. This is in essence the phenomenological correction (important primarily near saturation) that we have introduced to the HM ($N_{sp} \sim s^{1/d}$, $d = 2$ or $3 = \text{dimensionality}$). It postulates existence of cluster size or length scale at which individual populations would not interact within large aqueous habitats (e.g. separated by “diffusive spheres”). The exponent ($1/d$) suggests that the number of species per habitat grows with the average distance between any two points selected randomly within a single habitat of size s . Based on the reviewer comment, we have decided to use only one version the HM that allows multiple species per habitat throughout the manuscript. We discuss the difference in the two assumptions and compare the outcome in the supplementary materials (see Supplementary Figure 3 below)

R1.3: *Especially the question arises if Fig. 3 would look different allowing more than one species per habitat.*

We have tested and confirmed that changing the number of species per habitat would not alter the shape of the richness-water content relation considerably. It would stretch the curve towards larger values of richness as shown for two dimensions (surfaces) in comparison with simulation (SIM) results depicted in Supplementary Figure 2. Because we need to set a detection limit for comparing model predictions with observations a higher level of richness due to the increased number of species per habitat can be partially compensated for by selecting higher detection limits. The primary effect of including multiple species per habitat is manifested in the modeled species abundance distributions (Supplementary Figure 3) that affect evenness in Fig. 5 (see **R1.4**).

Supplementary Figure 2 | Comparison of the heuristic model (HM) with the spatially-explicit individual-based model (SIM) on surfaces (two dimensional domains). Square symbols and bars (mean \pm SD, $n = 12$) depict richness predicted by the SIM rarified to 1000 counts. The aqueous-phase fragmentation-based HM (solid line) captures the trend in simulated richness with water content ($\text{m}^3 \text{m}^{-3}$). The proportionality of the number of species per habitat N_{sp} to the domain's dimensionality (surface or volume) and to the size s of the aqueous habitats ($N_{sp} \sim s^{1/2}$) may explain the difference between SIM and the single species HM (dashed line).

Supplementary Figure 3 | Modeled and observed soil bacterial species abundance distributions (SAD). Comparison of relative abundances from empirical observations (x-axis) with estimates of the aqueous-phase fragmentation-based heuristic model (HM; y-axis). Scenarios with single and multiple species per aqueous habitat are compared to observations. A 1:1 line and Pearson correlations are shown for both soil bacterial diversity datasets. **a**, Relative SADs from the Earth Microbiome Project (EMP) and **b**, from a recent study by Delgado *et al.* (DEL) considering a single species per habitat. The consideration of multiple species per habitat with the number of species N_{sp} proportional to the dimensionality and size s of the habitat ($N_{sp} \sim s^{1/3}$) improves the agreement with model predictions for both datasets; **c**, EMP and **d**, DEL, respectively

R1.4: *Also in Fig. 5 it remains unclear how using the empirical input parameters changes the outcome and how this model change would affect the other results of the paper.*

We thank the reviewer for this insightful suggestion. The consideration of multiple species per habitat results in higher evenness with high water contents where only a few but large habitats emerge. As stated in the response above (**R1.2**), we now report only the multi species heuristic model (HM) and the agreement with the observations' central tendency improved considerably (median \pm IQR, Fig. 5). With the added flexibility, the modified HM (multiple species per aqueous habitat as a function of habitat size) is expected to perform better when comparing diversity metrics that consider species relative abundance directly (Supplementary Figure 3, see **R1.3**). Additionally, the evaluation of the HM for every sampled location circumvents the need to rely on empirical correlation of model inputs. Instead, we use the independent estimates of soil carrying capacity and climatic water contents for each sampled location and report the tendency of the HM predictions as a smoothed trend line (instead of using binned values; see Fig. 5 – solid line). Other results of the manuscript would not be affected (for example the predicted maps in Fig. 4 use independently estimated carrying capacity and climatic water contents while predictions for richness in Fig. 3 are based on median carrying capacity).

Fig. 5. Bacterial community evenness decreases with carrying capacity and climatic water contents. Evenness from two independent studies is shown together with estimated cell density (carrying capacity). Samples were aggregated by latitude, longitude and soil depth (EMP¹, n = 484 and DEL⁴, n = 218). The median and interquartile ranges (colored symbols and bars) are displayed for groups of water contents (bin width: 0.05, number of samples see Supplementary Table 2). Individual data points are shown for bins containing less than ten samples (small circles) and samples with cell density lower than 10¹² m⁻³ were removed. Evenness predicted by the heuristic model (HM) is calculated using paired values of climatic water content and carrying capacity (evaluated for every sample). Using the joint data of water content and cell density as model input, the HM reproduces the observed tendency of evenness. A locally weighted scatterplot smooth (LOWESS) of modeled evenness is shown for the HM predictions (solid line).

R1.5: *Fig2: Colors indicate 2-.5,50,97.5 percentile: That was quite confusing for me . The lines in the main plot correspond to specific temperatures (the percentiles) but the kernel densities correspond to temperature ranges. So I wonder how the two sets of curves are comparable and if they should have the same colors?*

We thank the reviewer for pointing out this ambiguity and the lack of clarity in the original figure. We adapted the figure so that the curves represent the medians of three groups of mean annual temperature (MAT). The binning was adapted to represent values of MAT < 0 °C, from 0 to 15 °C, and > 15 °C. Each group's median is then used as input for each curve. The range of cell densities within each bin is shown as boxplots for each class of MAT (with box and bars representing the central 50 and 95% of values, respectively).

Fig. 2. Soil bacterial abundance varies in relation to net primary productivity (NPP), mean annual temperature (MAT) and soil depth. **a**, Bacterial cell density at soil carrying capacity as a function of NPP with model estimates sensitive to MAT (solid lines). Estimates are compared with measured data of microbial biomass¹ converted to bacterial cell density and are grouped by temperature (MAT ≤ 0 °C, 0 °C $<$ MAT ≤ 15 °C, MAT > 15 °C). Each group's median is reported in the figure legend in blue, green and orange, respectively. The distributions of cell densities are indicated for each temperature group as the central 50 and 95% range. **b**, Variations of bacterial cell density with soil depth. The log-normal fit provides bounds on cell density (carrying capacity) for intermediate MAT (solid line) and for the central 95% of NPP (shaded area). Observed estimates of cell density are reported for their average sampling depth. Most samples were taken above 10 cm as shown in the boxplot.

R1.6: *Fig. 5: For me its hard to see a decrease of evenness in the data. I think if the authors believe there is a decrease they should give a slope of a line fit and its error or similar.*

Indeed the observed decrease in evenness is relatively small, as evidenced by the low, yet negative Pearson correlation coefficients relating evenness to climatic water contents for samples from the two datasets (-0.17 and -0.41 for EMP and DEL, respectively: Supplementary Figure 5a).

Mechanistically, we expect an overall decrease in soil bacterial evenness as the soil becomes wet. This is predicted by both the HM and the SIM independently. We note however, that pre-processing of relative abundance information (e.g. removal of singletons) can significantly alter the apparent relation as demonstrated with the SIM (Supplementary Figure 8b). Hence, we do not necessarily expect to observe a monotonous decrease. Nonetheless we fit a linear model of the form: $Evenness \sim \alpha + \beta\theta_{\tau} + \gamma\log(\rho_{cell})$; with climatic water content (θ_{τ}) and cell density (ρ_{cell}). Although, the goodness of fit is not large ($R_{adj}^2 = 0.14$), both slopes are negative and the magnitude of the intercept appears reasonable ($\alpha = 1.06 \pm 0.09$; $\beta = -0.31 \pm 0.08$; $\gamma = -0.05 \pm 0.01$). Additionally, the residuals indicate no model bias (Supplementary Figure 5b).

Supplementary Figure 5 | Empirically observed trends of soil bacterial evenness. **a**, Decrease of bacterial community evenness with climatic water content. Pearson correlation r for individual samples of both diversity datasets (EMP¹ $n = 2871$, DEL⁴ $n = 237$) are indicated in the legend. The trend line shows a linear model (LM, see **b**) evaluated for median cell densities. **b**, The linear model was fitted to the empirical data for all sampled locations ($n = 684$) and the response surface of evenness is shown as a function of climatic water contents and cell densities (colored contours). Samples with cell densities lower than 10^{12} m^{-3} were removed prior to fitting the model as indicated in the figure. Negative slopes (β , γ) suggest that evenness is jointly reduced by increasing climatic water contents and cell density. Model residuals are not indicative of a persistent bias. Additionally, evenness is shown for bins of water contents (median \pm IQR) to highlight the central tendency.

Some suggestions about the figures (that can be ignored):

R1.7: *Maybe use the word ‘model’ or similar in the title to manage expectations a little*

We included the term ‘modeled’ to label figure axis where applicable.

R1.8: *Fig1: maybe show more similar looking species in right panel of soil comic.*

Species in the right panel (wet and connected soil) have been modified to appear more similar.

R1.9: *Fig2: maybe color datapoints according to temperature range*

We tried coloring data points by temperature range but decided to not keep the color as many points overlap (as evidenced by the distributions of cell density values) and little additional information could be displayed. Nonetheless, we adapted the presentation of the figure to show more clearly the cell densities to expect under different ranges of temperature (see **R1.5**).

R1.10: *Fig3: Line 74 estimates of soil pH: how was it estimated? Measured?*

We wrongfully cited SoilGrids⁴ (global digital soil maps) as the source of soil pH estimates, which was the case in an earlier version of the manuscript. The current values of soil pH were reported by Delgado-Baquerizo *et al.* and originate from sample scale measurements³.

I617: *“Colors indicate reported soil pH, which has been shown to be affected by climate”*

R1.11: *Fig5: maybe label SIM and HM symbols according to cell density*

The calculations of the HM in figure 5 have been modified. We evaluated the HM with paired climatic water content and cell density estimates for each sample and display a smoothed trend line of the resulting evenness (see **R1.4**). It is therefore not possible to label the symbols by cell density. However, cell densities of the SIM are reported in Supplementary Figure 7.

Supplementary Figure 7 | Comparison of evenness estimated using the aqueous-phase fragmentation-based heuristic model (HM) and the spatially-explicit individual-based model (SIM) for different water contents and carrying capacity. The HM (solid line) is evaluated in two dimensions for every value pair of modeled cell density and water contents obtained from the SIM (square symbols and bars – mean \pm SD, $n = 12$). Colors indicate modelled cell density from the SIM.

Reviewer #3 (Remarks to the Author):

R3.1: *I found the manuscript to be much improved and much more readable and accessible than the original. I think the authors do a good job showing that this general approach is relevant to predicting patterns of microbial diversity. They also do a better job of discussing the limitations and assumptions of their approach.*

We thank the reviewer for the encouraging comment and are happy that improvement could be noticed regarding our discussion of limitations and assumptions.

R3.2: *I think especially the abstract could still do a better job of zooming in on the question at hand, so as to capture the attention of the reader and clearly guide him/her to what is going to be addressed in the manuscript. For instance, the first sentence is so general as to not be very useful. This could be stated much more sharply. Perhaps something like “microbial diversity has been shown to vary across terrestrial habitats, with presumptive links to function.” Also, it would be helpful to clearly state the question that is being addressed.*

We agree that the abstract could be formulated more concisely. The suggested phrase was adopted and we included an explicit statement of what is addressed in the manuscript (“Here we [...]”). The abstract has been revised to better reflect the content of the manuscript.

Herewith some specific comments as I went back through the manuscript:

R3.3: 1.) *Introduction: the structure of the introduction would be improved by the use of paragraphs.*

We followed the suggestion and used paragraphs where applicable.

R3.4: 2.) *L21: some studies suggest even higher numbers – might be worth including.*

A study⁵ reporting soil bacterial diversity in the order of 10^6 was included. We also compared to a recent study⁶ on the expected number of phylotypes on earth to provide an upper bound on what could be expected in terrestrial environments.

I26: “The number of bacterial phylotypes ranges between 10^2 to 10^6 per gram of soil^{2,3,7}, with high values similar to the richness in all of earths environments⁶.”

R3.5: 3.) *L24: I think it would be better to refer to “the rare components of the soil microbiome”*

We thank the reviewer for this suggestion. The phrase was adopted and improved the flow of the text.

I30: “This wide range of microhabitats is particularly important for maintaining the rare components of the soil microbiome.”

R3.6: 4.) *L62 (and elsewhere) – I think it is important to make it clear that you refer to terrestrial biomes – also, I think this sentence should be rearranged: thus... Modeled trends of soil bacterial carrying capacity and diversity were compared to empirical observations across different terrestrial biomes”*

The sentence was rearranged as recommended, and we specified (where applicable) that the biomes considered are terrestrial.

I70: “Modeled trends of soil bacterial carrying capacity and diversity are compared to empirical observations¹⁻³ across terrestrial biomes.”

R3.7: 5.) *L80-82L I suggest switching the order around of this sentence – we found that varying the range of expected values had little impact on carrying capacity estimates. We therefore used a constant value in our model.*

We followed the suggestion that greatly improves the logical structure of the sentence.

I88: “We found that varying the range of expected values (14-30% of NPP⁸) had little impact on estimates of carrying capacity. A constant value of this respiratory fraction was therefore considered based on mechanistic model simulations⁸.”

R3.8: 6.) L202: *I think you should define what you mean by hotspots – hot spots for what? Activity? Interaction? Diversity?*

The ambiguous usage of the term “hot-spots” was avoided. We now explicitly refer to “nutrient hot-spots” and with that implicitly to potential aspects for bacterial activity, interactions and diversity.

I228: “This could be due to dominance of a few species that may cluster around nutrient hot-spots⁹, or loss of oligotrophic species that would be outcompeted in well-connected and dense communities.”

R3.9: 7.) L223-224: *This could of course also be approached experimentally.*

We agree with the reviewer that this could be approached experimentally. We would further suggest that experimental validation would be essential in disentangling the effects of soil carbon and water on bacterial habitats and included a corresponding statement.

I250: “Teasing apart such confounding associations requires detailed statistical analysis and experimental validation, which are best conducted in dedicated studies”

R3.10: 8.) L247: *On the more positive side, you might also mention restoration efforts (as opposed to only aspects that cause soil degradation).*

Following the suggestion we included restoration efforts as part of changes in land use.

I275: “[...](e.g. in intensity of agricultural management or restoration to natural ecosystems)[...]”

R3.11: 9.) Figure 1: *I think the drawing might be improved by also included the soil matrix – it is not clear what the white background depicts. L550; later you say that his assumption does not hold, so I think this has to be toned down here. Also, I think that you can make a better distinction between potential carrying capacity and realized carrying capacity.*

We thank the reviewer for these helpful suggestions. The soil matrix was included in the illustration and the white background was removed (Fig 1). We toned down the wording in the figure caption to indicate the possibility of having multiple species in a single aqueous habitat (I589). Additionally, a sentence was added to clarify the distinction between potential and realized carrying capacity (I591).

I589: “When the soil becomes sufficiently dry almost all aqueous habitats are physically isolated and might contain only a few species.”

I591: “The specific carrying capacity in a biome is based on carbon input flux and temperature that establish an upper bound on bacterial cell density (rarely realized in any particular location due to other limiting factors).”

R3.12: 10.) *Figure 3: Empirical data is extremely sparse at the dry end of the spectrum (zero and one point in panels a and respectively) – thus the “real” data not show strong support for the sharp diversity decline under the driest conditions here. I think this deserves mention.*

We thank the reviewer for pointing out the lack of discussion regarding the sample coverage at the dry end. This also led us to discover a mistake in the presentation of the data, which removed two data points. Additionally, we adapted the binning of data to comply with *Nature Communications* guidelines of displaying average values only if the number of samples is greater than ten. Wherever there are less samples, the individual data points are now shown. To further improve the representation we newly grouped samples of the EMP dataset by top- and sub-soil (<25cm and ≥25cm). Nonetheless, we now also address the sparsity of data used at the dry end explicitly in the discussion (I209). We recently published a global, statistical meta-analysis¹⁰ with an increased number of sampled locations in dry regions that exhibit a drop in diversity under low climatic water contents at larger scales; as previously reported for increased aridity¹¹. Furthermore, a sharp decline in diversity under very dry conditions was also reported for dry valleys of Antarctica¹² and a drop in richness (Faith's PD) and Shannon index with soil relative humidity was observed in the Atacama desert¹³. A distinct drop in richness towards dry and wet conditions was also observed in soil microcosm experiments when rare species were emphasized¹⁴. The studies mentioned were included in the discussion section. However, we could only speculate about other factors that might cause the lack of clear patterns, particularly the absence of residual soil moisture in the HM (that could make water contents appear too low) and the possible influence of dew (that could enhance bacterial growth) in some dry regions.

I209: “The data available at low climatic water contents are sparse and do not provide support for the predicted steep decline of bacterial diversity as soil becomes dry that was previously reported with increased aridity at large scales¹⁴. However, a significant decrease in bacterial richness was also observed in a recent statistical meta-analysis for climatic scales³⁰ and could be confirmed using the SIM (Fig. 3b). Additionally, it has been reported that bacterial diversity declines sharply with moisture in dry soils of Antarctica²² and decreases with soil relative humidity along transects of the Atacama desert³¹. Microcosm experiments revealed an increase in richness with moisture that peaks at intermediate water contents that promote rare bacterial species³².”

R3.13: Minor corrections/suggestions:

We thank the reviewer for the detailed corrections and suggestions. We incorporated all changes as suggested below.

L13: change “vary” to “varies”

Amended.

L32; change “for” to “toward”

Amended.

L37: change “deciphering” to “understanding”

Amended.

L54: delete “diverse” - delete “The” from next sentence

Amended.

L97: delete; “In the following”

Amended.

L106: change “additionally” to “also”

Amended.

L109: add commas after “dataset” and “depth”

Amended.

L114: change to read, “... data, but rather are...”

Amended.

L125: change “shift’ to “discrepancy” and change “is” to “can be”

Amended.

L131: change “to” to “for each”

Amended.

L137: change “thus’ to “therefore”

Amended.

L149: add “patterns” after “diversity”

Amended.

L151: change “exhibits” to “exhibit”

Amended.

L152: change “and appear” to read “, with this patterns being”

Amended.

L159: insert “with” before “carrying” – and change next sentence to start “These results are...”

Amended.

L177: change “has shown” to “shows”

Amended.

L191: I suggest using the word “realize” instead of “express”

Amended.

L197: change “seem” to “seem”

Amended.

L232: change “inhabit” to “comprise”

Amended.

L555: change to read “Soil bacterial abundance in relation to net...”

Amended.

L592: delete “is”

Amended.

References

1. Xu, X., Thornton, P. E. & Post, W. M. A global analysis of soil microbial biomass carbon, nitrogen and phosphorus in terrestrial ecosystems. *Global Ecology and Biogeography* **22**, 737–749 (2013).
2. Thompson, L. R. *et al.* A communal catalogue reveals Earth’s multiscale microbial diversity. *Nature* **551**, 457–463 (2017).
3. Delgado-Baquerizo, M. *et al.* A global atlas of the dominant bacteria found in soil. *Science* **359**, 320–325 (2018).
4. Hengl, T. *et al.* SoilGrids250m: Global gridded soil information based on machine learning. *PLoS one* **12**, e0169748 (2017).
5. Gans, J., Wolinsky, M. & Dunbar, J. Computational Improvements Reveal Great Bacterial Diversity and High Metal Toxicity in Soil. *Science* **309**, 1387–1390 (2005).
6. Louca, S., Mazel, F., Doebeli, M. & Parfrey, L. W. A census-based estimate of Earth’s bacterial and archaeal diversity. *PLOS Biology* **17**, e3000106 (2019).
7. Bahram, M. *et al.* Structure and function of the global topsoil microbiome. *Nature* **560**, 233–237 (2018).
8. Fatichi, S., Manzoni, S., Or, D. & Paschalis, A. A mechanistic model of microbially mediated soil biogeochemical processes - a reality check. *Global Biogeochemical Cycles* (2019) doi:10.1029/2018GB006077.
9. Nunan, N., Leloup, J., Ruamps, L. S., Pouteau, V. & Chenu, C. Effects of habitat constraints on soil microbial community function. *Scientific Reports* **7**, (2017).
10. Bickel, S., Chen, X., Papritz, A. & Or, D. A hierarchy of environmental covariates control the global biogeography of soil bacterial richness. *Sci Rep* **9**, 1–10 (2019).
11. Maestre, F. T. *et al.* Increasing aridity reduces soil microbial diversity and abundance in global drylands. *PNAS* **112**, 15684–15689 (2015).
12. Lee, K. C. *et al.* Stochastic and Deterministic Effects of a Moisture Gradient on Soil Microbial Communities in the McMurdo Dry Valleys of Antarctica. *Front. Microbiol.* **9**, 2619 (2018).
13. Neilson, J. W. *et al.* Significant Impacts of Increasing Aridity on the Arid Soil Microbiome. *mSystems* **2**, (2017).
14. Banerjee, S. *et al.* Legacy effects of soil moisture on microbial community structure and N₂O emissions. *Soil Biology and Biochemistry* **95**, 40–50 (2016).

Reviewers' Comments:

Reviewer #1:

Remarks to the Author:

The authors addressed all my concerns. From my perspective the manuscript is ready to be published.
I congratulate the authors for this nice work.

Reviewers' comments:

Reviewer #1 (Remarks to the Author):

The authors addressed all my concerns. From my perspective the manuscript is ready to be published. I congratulate the authors for this nice work.

We thank the reviewer for the insightful comments and helpful suggestions during the peer review process.